# CAN IN-CONTEXT LEARNING REALLY GENERALIZE TO OUT-OF-DISTRIBUTION TASKS?

**Qixun Wang[1]**     **Yifei Wang[2]**     **Xianghua Ying[1,3]\***     **Yisen Wang[1,3]\***

[1] State Key Lab of General Artificial Intelligence,
   School of Intelligence Science and Technology, Peking University
[2] MIT CSAIL
[3] Institute for Artificial Intelligence, Peking University

## ABSTRACT

In this work, we investigate the mechanism of in-context learning (ICL) on out-of-distribution (OOD) tasks that were not encountered during training. To this end, we conduct synthetic experiments using a GPT-2 model to learn OOD mathematical functions through ICL. Our findings reveal that Transformers may struggle to learn OOD tasks via ICL. Specifically, ICL operates by selecting a function within the pretraining hypothesis space and optimizing it via gradient descent using in-context examples, rather than learning truly novel functions. Additionally, we examine ICL's well-documented ability to infer unseen abstract labels in context. We demonstrate that this ability only holds in scenarios without distributional shifts, suggesting that it does not constitute genuine new-task learning. Furthermore, we analyze ICL's OOD performance when pretrained on multiple tasks. Both empirical and theoretical results reveal a low-test-error preference, where ICL tends to select the pretraining function that minimizes test error rather than adapting to entirely new tasks. We validate this phenomenon through numerical experiments. Our theoretical insights, combined with empirical findings, provide a deeper understanding of ICL's limitations and its underlying mechanism when tackling OOD tasks. Code is available at `https://github.com/NOVAglow646/ICL-OOD`.

## 1 INTRODUCTION

Pretrained large language models (LLMs) can perform in-context learning (ICL) (Brown, 2020), where providing a few examples of input-output pairs and a query example improves the model's ability to generate the desired output, compared to zero-shot predictions. Understanding ICL's ability to learn out-of-distribution (OOD) input-output relationships, which are unseen during training, has recently gained significant attention.

Recent studies have demonstrated that ICL can tackle seemingly new tasks. For instance, Garg et al. (2022); Raventós et al. (2023); Zhang et al. (2023a); Akyürek et al. (2023) found that ICL can learn new linear regression weights after pretraining on a large set of weight vectors. Moreover, Pan (2023); Kossen et al. (2024); Vacareanu et al. (2024) revealed that real-world LLMs like Llama-2 (Touvron et al., 2023) and GPT-4 (Achiam et al., 2023) are capable of solving artificially constructed tasks likely unseen in their pretraining data, such as a classification task with abstract labels.

However, another line of research (Yadlowsky et al., 2023; Ahuja & Lopez-Paz, 2023) has raised a contrasting view, showing that ICL struggles to generalize to OOD tasks where there are distributional shifts in either the input distribution $P(X)$ or the input-label mapping $P(Y|X)$. These findings raise several important questions:

*Can ICL really learn new input-output mappings from the context?*
*What underlying mechanism of ICL determines its performance on OOD tasks?*

---

\*Corresponding Authors.

This work aims to consolidate previous findings by addressing these questions. First, we empirically show that when pretrained on a specific function class, the OOD performance of ICL approaches that of a model from the same function class optimized via gradient descent. This suggests that ICL tends to implement the function class encountered during pretraining. Additionally, we investigate the widely observed phenomenon where ICL can classify abstract unseen labels. We find that the ability to solve such a task is a form of retrieval capability, which disappears when faced with OOD classification rules, indicating that success in these tasks does not necessarily reflect an inherent ability to learn new tasks. Finally, we explore scenarios in which the model is pretrained on multiple tasks, empirically uncovering the algorithm selection mechanism for OOD tasks. We also theoretically reveal the algorithm-selection mechanism for ICL. Our main contributions are:

1. We empirically show that ICL tends to implement the pretraining function based on the downstream task context, highlighting its limitation in solving OOD tasks (Section 3.1).

2. We further investigate ICL's ability to classify unseen abstract labels. We find that such tasks can be solved by retrieving similar examples from the context. This retrieval ability can arise from training on tasks with more diverse abstract labels (Section 4.1) and only emerges when the test function is in distribution (Section 4.2). We further validate our findings by showing that pretrained Llama-3-8B (Dubey et al., 2024) and Llama-2-7B fail to learn OOD functions through ICL in a synthetic vector classification task (Section 4.3).

3. We explore the ICL's behavior when trained on multiple tasks, and observe that the algorithm selection mechanism broadly occurs in OOD scenarios. We theoretically prove the **low-test-error** preference of ICL prediction, i.e., the ICL prediction prefers to implement the pretraining function with low test error (Section 5.1). We also validate our theory with numerical experiments (5.2).

## 2 Existing Theoretical Predictions of ICL

Previous literature has provided some theoretical insights into the behavior of ICL. Here we briefly review some of the representative findings. 1) **ICL makes Bayesian predictions.** Xie et al. (2022); Wies et al. (2024); Zhang et al. (2023b) theoretically demonstrated that ICL behaves like a Bayesian-optimal predictor, i.e., it will infer a task concept based on the given test context, and then predict using the inferred task and the input prompt. However, these Bayesian frameworks do not depict the concrete process of how the task concept is inferred, especially for OOD scenarios. 2) **ICL implements gradient descent (GD).** Von Oswald et al. (2023); Zheng et al. (2024) construct specific Transformer weights on which the ICL prediction is equivalent to a linear regression predictor or alignment objective (Wang et al., 2024a) optimized by gradient descent. 3) **ICL implements algorithm selection.** Bai et al. (2023); Wang et al. (2024b) demonstrate the existence of Transformers that can realize algorithm selection between linear classification and regression by constructing specific Transformer weights. 4) **ICL performs retrieval.** Li et al. (2024) proves that a trained non-linear Transformer will concentrate its attention of the query on the in-context examples possessing similar features to that of the query.

These theoretical findings may seem disparate, as they describe different aspects of ICL under varying assumptions and settings. Furthermore, many of them rely on oversimplified model architectures or deliberately constructed model weights to reach their conclusions, limiting their practical applicability. In the following sections, we aim to provide a unified perspective on ICL by conducting experiments with deep nonlinear Transformers on both synthetic and real-world OOD tasks.

## 3 Exploring the Performance of ICL on OOD Tasks

### 3.1 GPT-2 Implements the Functions Class Seen During ICL Pretraining

**Evaluating GPT-2 on unseen mathematical function classes.** To investigate the ICL performance on new tasks that are unseen during training, following Garg et al. (2022), we train a GPT-2 (Radford et al., 2019) from scratch on some simple functions and evaluate it on functions different from the training ones. Denote the Transformer model parameterized by $\theta$ as $M_\theta$. The pretraining objective is $\min_\theta \frac{1}{T} \sum_{i=1}^{T} \mathbb{E}_{f \sim \mathcal{F}}[\|M_\theta(\mathcal{S}_i \oplus \boldsymbol{x}_{i+1}) - f(\boldsymbol{x}_{i+1})\|_2^2]$, where $\mathcal{S}_i = [\boldsymbol{x}_1 \oplus y_1 \oplus \boldsymbol{x}_2 \oplus y_2 \oplus ... \oplus \boldsymbol{x}_i \oplus y_i] \in$

$\mathbb{R}^{d \times 2i}$ is the context, $\oplus$ denotes concatenation. $\boldsymbol{x}_i \in \mathbb{R}^d$ are sampled from a standard Gaussian distribution $\mathcal{N}(0,1)$ with dimension $d = 20$. Let $y_i = f(\boldsymbol{x}_i) \in \mathbb{R}$ represent the labels, with $\mathcal{F}$ denoting the hypothesis class to which $f$ belongs. We train a GPT-2 model on one of the three function classes $\mathcal{F}$: linear regression (LR), quadratic regression (QR), and a 2-layer ReLU neural network (ReLU NN, detailed descriptions in Appendix C.1). We then evaluate its ICL performance on all three function classes. For comparison, we also train models of the corresponding $\mathcal{F}$ with gradient descent (GD) using the test in-context examples (details in Appendix C.1).

**Observations.** We plot the test error on the three tasks in Figure 1 and observe that: 1) when evaluated on the same task $\mathcal{F}$ as pretraining, ICL can reach near-zero test error, which is consistent with the findings in Garg et al. (2022). 2) when evaluated on a new task, ICL performs similarly to the corresponding model of the pretraining function class optimized by GD given enough in-context examples. This indicates that the ICL prediction implements the training function classes. 3) The models trained on linear and quadratic regression exhibit a double descent error curve (Nakkiran, 2019), characterized by a high error when given exact $d$ examples and evaluated on a new task. This further demonstrates that ICL implements in-distribution (ID) predictions, as the double descent curve is a distinctive phenomenon unique to linear regression models.

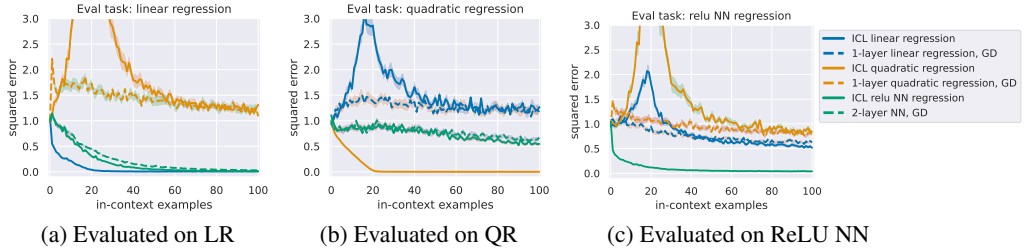

(a) Evaluated on LR  (b) Evaluated on QR  (c) Evaluated on ReLU NN

Figure 1: The ICL test error of Transformers trained on different function classes (solid lines) and the performance of models from the corresponding pretraining functions classes trained by GD using the test in-context examples (dashed lines). Y-axis: test square error. X-axis: context length. We observe that as the test context length increases, the ICL performance pretrained on a particular function class closely approaches that of the model from this function class trained by GD.

## 3.2 REAL-WORLD LLMS TEND TO MAKE IN-DISTRIBUTION PREDICTIONS DURING ICL

In this section, we will demonstrate how the tendency of ICL to perform ID predictions manifests in real-world LLMs. We designed a task involving label prediction with reversed letters (e.g., in sentimental classification, "positive"→"evitisop"). We found that in this task, a pretrained Llama-3-8B (Dubey et al., 2024) prefers to output the inversion of the query word rather than predict the reversed correct label, as shown in Figure 2. Although both reversal tasks are rare, directly outputting the reversed version of a word is more common than first reasoning and then outputting the reversed prediction. This result suggests that LLMs, when performing ICL, are more inclined to implement ID tasks. For more details, refer to Appendix C.2. Inspired by Raventós et al. (2023), We also explore whether increasing the diversity of training tasks, while keeping them ID, can activate the OOD generalization ability. The results in Appendix B.4 also suggest a negative conclusion.

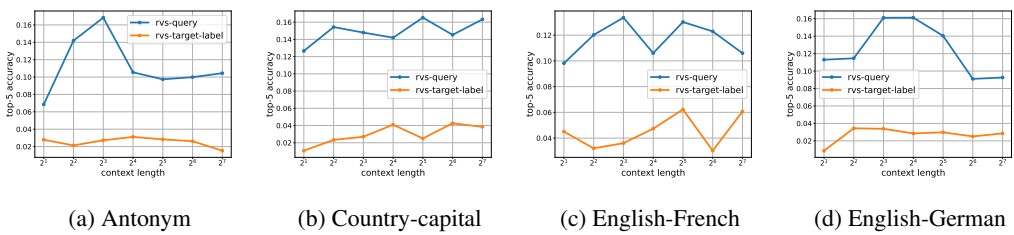

(a) Antonym  (b) Country-capital  (c) English-French  (d) English-German

Figure 2: The top-1 accuracy of predicting the reversed query word (blue) and predicting the reversed target label word (orange). The accuracy of predicting the reversed query word is higher than outputting the reversed target, indicating ICL makes ID predictions.

**Summary of the Empirical Results & Connections with the Existing Theories (I)**

Given an OOD context, ICL finds a near-optimal solution within its pretraining task space. Particularly, when learning OOD mathematical functions, ICL behaves as a predictor of its pretraining function class optimized by GD using the in-context examples. This validates and extends existing results by Zhang et al. (2023a) which theoretically shows that linear attention models trained on linear regression data still implement linear regression given arbitrary downstream context (see Appendix D).

## 4 LEARNING ABSTRACT LABELS MAY NOT BE A REAL OOD CAPABILITY

### 4.1 CLASSIFYING ABSTRACT LABELS IS A PREDICT-THEN-RETRIEVE PROCESS THAT CAN EMERGE FROM TRAINING

Recent works (Pan, 2023; Kossen et al., 2024) have shown that LLMs can successfully perform classification tasks in which the labels are "abstract symbols" with no semantic meaning (e.g., in sentimental classification, "positive" and "negative" are replaced with "foo" and "bar", respectively). These tasks are likely not encountered during pretraining. Pan (2023) refer to this ability of ICL to perform such classification as "task learning" (TL). In this section, we explore whether this really reflects a new-task-learning capability of ICL, or if it is something else.

**Training GPT-2 to perform a retrieval task through ICL.** The classification of abstract labels can be approached by retrieving an example with semantics similar to the query and then outputting the label of that example. Therefore, the retrieval ability is a crucial prerequisite for performing abstract-label classification. We design a retrieval task to investigate whether ICL's retrieval capability can emerge from training. Specifically, we generate a predefined word embedding $E \in \mathbb{R}^{N \times d}$ and randomly sample $\boldsymbol{x}_i \in \mathbb{R}^d$ from the first 5 rows of $E$. Suppose vector $\boldsymbol{x}_i$ is the $I_{\boldsymbol{x}_i}$-th row of $E$, i.e., $\boldsymbol{x}_i = E_{I_{\boldsymbol{x}_i}}$. To generate the labels $\boldsymbol{y}_i \in \mathbb{R}^d$, we first map the index $I_{\boldsymbol{x}_i}$ to $I_{\boldsymbol{y}_i} \in [N]$ using the mapping rule $I_{\boldsymbol{y}_i} = I_{\boldsymbol{x}_i} + s$, where $s \in \mathbb{N}$ is randomly sampled. Next, we set $\boldsymbol{y}_i = E_{I_{\boldsymbol{y}_i}}$. All in-context examples in a sequence share the same mapping rule defined by $s$. To succeed in this task, the model must retrieve the same token as the query example from the context and output its subsequent token. All models are trained with 200,000×64 sequences, where 200,000 is the number of training steps and 64 is the batch size. We train three models with three different ranges of $s$: $s \sim \mathcal{U}(50, 150)$, $s \sim \mathcal{U}(50, 250)$, and $s \sim \mathcal{U}(50, 450)$ and evaluate on $s \sim \mathcal{U}(50, 150)$, $s \sim \mathcal{U}(10, 20)$, and $s \sim \mathcal{U}(500, 600)$, where $\mathcal{U}$ denotes the uniform distribution.

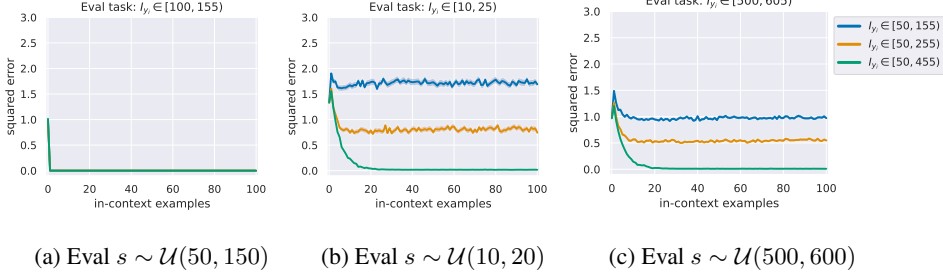

(a) Eval $s \sim \mathcal{U}(50, 150)$     (b) Eval $s \sim \mathcal{U}(10, 20)$     (c) Eval $s \sim \mathcal{U}(500, 600)$

Figure 3: The ICL test error of Transformers trained on the retrieval task with different numbers of label tokens. "Eval" denotes "evaluated on". The ability to retrieve OOD labels emerges from increasing the number of training mappings (larger range of $s$).

**Observations.** In Figure 3, all three models perform well when labels are ID (a). When the labels are OOD, the ICL performance improves with the number of label vectors (random mappings) encountered during training (b, c). This demonstrates that the ability to retrieve arbitrary labels from the context can emerge from training on diverse retrieval tasks. These findings may also offer new insights into how real-world LLMs develop in-context retrieval capabilities: when autoregressive pretraining includes numerous instances requiring the model to retrieve tokens from previous contexts, such abilities can emerge. We further validate this finding by observing the emergence of induction heads in Appendix B.1.

**Training GPT-2 to perform a predict-then-retrieve task through ICL.** To further explore the emergence of the ability to classify abstract labels, we design a predict-then-retrieve task that emulates the natural language classification with abstract labels. In this task, $\boldsymbol{y}_i = E_{I_{\boldsymbol{x}_i}}$, where $I_{\boldsymbol{x}_i} = \text{floor}(0.4 * (\boldsymbol{w}^\top \boldsymbol{x}_i)) + s$, with $E$ being the predefined word embedding and $s \in \mathbb{N}+$ shared in a ICL sequence. Here, $\boldsymbol{x}_i, \boldsymbol{w} \sim \mathcal{N}(0,1) \in \mathbb{R}^d$. In this task, estimating $\boldsymbol{w}$ and calculating $\boldsymbol{w}^\top \boldsymbol{x}_i$ simulates predicting the original label ("positive" and "negative") by in-context learning the natural language task, while retrieving $\boldsymbol{y}_i$ from $\boldsymbol{x}_i$ such that $\text{floor}(0.4 * (\boldsymbol{w}^\top \boldsymbol{x}_i)) = \text{floor}(0.4 * (\boldsymbol{w}^\top \boldsymbol{x}_{query}))$ resembles identifying the abstract labels ("foo" and "bar")[1], where $\boldsymbol{x}_{query}$ is the query example. Again, we train three models on different ranges of mappings: $s \sim \mathcal{U}(100, 200)$, $s \sim \mathcal{U}(100, 1000)$, and $s \sim \mathcal{U}(100, 2000)$, and evaluate on $s \sim \mathcal{U}(100, 200)$, $s \sim \mathcal{U}(500, 600)$, and $s \sim \mathcal{U}(3000, 3100)$.

**Observations.** In Figure 4, the ability to predict and retrieve unseen labels also improves as the number of labels encountered during training increases. This suggests that as long as the LLM has been exposed to sufficiently many similar tasks during training, it can effectively classify arbitrary OOD labels retrievable from context through ICL.

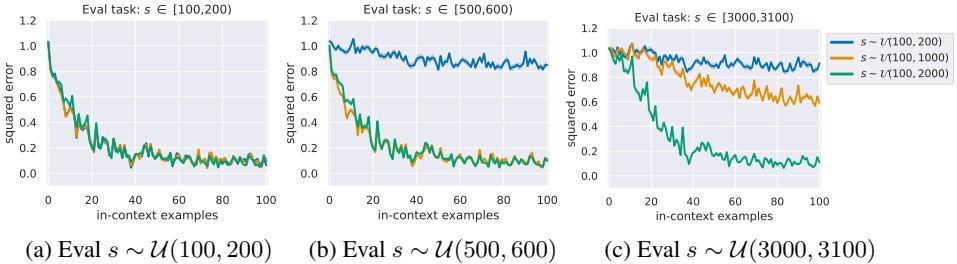

(a) Eval $s \sim \mathcal{U}(100, 200)$   (b) Eval $s \sim \mathcal{U}(500, 600)$   (c) Eval $s \sim \mathcal{U}(3000, 3100)$

Figure 4: The ICL test error of Transformers trained and tested on the linear regression + retrieval task with different numbers of label tokens. "Eval" denotes "evaluated on". The performance on unseen labels increases with the number of labels seen during training. Note that in this experiment, the underlying test function class is still ID.

## 4.2   ABSTRACT LABEL CLASSIFICATION CAN ONLY BE ACHIEVED ON ID TASKS

**A predict-then-retrieve task with OOD testing functions.** Given the above observations, one might question whether, once the target labels appear in the context, ICL can generalize beyond the training function class by retrieving the target label from the context. To investigate this, we conduct the same predict-then-retrieval task as in Figure 4 but replace the test functions with quadratic regression while preserving linear regression as the pretraining task.

**Observations.** The results in Figure 5 show that the generalization does not improve with training on more ID functions, even when the target label vectors appear in the context. Combining observations from Figure 4, we conclude that ICL can only solve classification with unseen labels over *ID* test function classes. This finding highlights a limitation in improving an LLM's performance through in-context examples. While providing examples with shared labels may seem helpful, this approach may fail if the underlying prediction rule is too OOD for the LLM to learn.

---

**Summary of the Empirical Results & Connections with the Existing Theories (II)**

To handle classification with abstract labels, the model infers an input-label mapping to implicitly establish a similarity metric. It then retrieves in-context examples similar to the query to deduce the OOD labels using this metric. However, this process succeeds only when the underlying function class is ID, thus it does not represent a true OOD generalization capability. This observation aligns with Bayesian frameworks for ICL—the implicit similarity metric here corresponds to the task concept inferred by the model. We leave an intuitive Bayesian interpretation of the findings in Section 4.1 and 4.2 to Appendix A.4.

---

[1]In our experimental setup, given a sufficiently long context ($\approx 50$), the label of the query is highly likely to appear in the context, as the number of the possible classes is far less than the number of in-context examples.

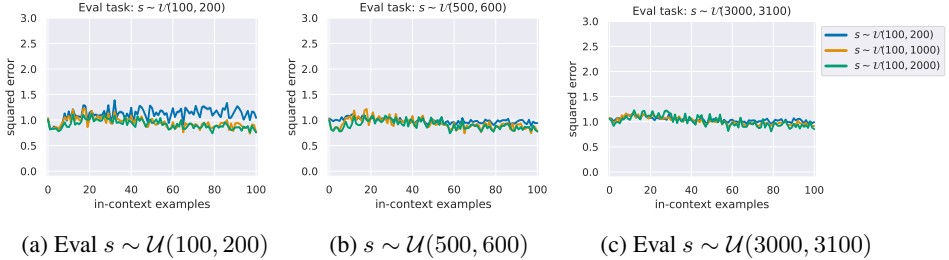

(a) Eval $s \sim \mathcal{U}(100, 200)$     (b) $s \sim \mathcal{U}(500, 600)$     (c) Eval $s \sim \mathcal{U}(3000, 3100)$

Figure 5: The ICL test error of Transformers evaluated on a quadratic regression + retrieval task. Different colors denote models trained on the linear regression + retrieval task with different numbers of label tokens. "Eval" denotes "evaluation". The model trained on $s \sim \mathcal{U}(100, 2000)$ doesn't generalize better than the other two models.

### 4.3 REAL-WORLD LLMS MAY NOT NECESSARILY IN-CONTEXT LEARN NEW TASKS

**Evaluating Llama-3 on an OOD synthetic vector classification task.** Now we assess whether real-world LLMs can tackle OOD tasks through ICL. We design a synthetic vector classification task for a pretrained Llama-3-8B. Specifically, we randomly sample $\boldsymbol{x}_i \in \mathbb{R}^d$ from the word embedding of Llama-3-8B (denoted as $E_{llama}$) and generate random linear mappings $\boldsymbol{W} \in \mathbb{R}^{d \times C}$ as task functions (where $C = 10$). The label vectors are created by mapping $\boldsymbol{x}_i$ to one of the ten label vectors in $E_{llama}$ using $\boldsymbol{W}$. Experimental details are in Appendix C.4. To complete this task, the model must learn $\boldsymbol{W}$ in context, which is unlikely to have been seen during the pretraining of Llama.

For comparison, we also evaluate the ICL performance of Llama-3-8B on a retrieval version of this task. Concretely, we first randomly sample different vectors from $E_{llama}$ as $\boldsymbol{x}_i$ and compute $\boldsymbol{y}_i$ in the same way as the above classification task to get $S = [\boldsymbol{x}_1, \boldsymbol{y}_1, ..., \boldsymbol{x}_C, \boldsymbol{y}_C]$ ($C = 10$). Then we repeat $S$ for 20 times to construct the input sequence $S' = [S \oplus S \oplus ... \oplus S]$, where $\oplus$ denotes concatenation operation. The goal is to predict the next token given a prefix of $S'$. To succeed in this task, the model has to retrieve the same token as the query token from the context and output its subsequent token $\boldsymbol{y}_i$.

**Observations.** The results of these two tasks are presented in Figure 6. We observe that the ICL performance on the synthetic classification task is close to random guessing (10% accuracy), while performance on the retrieval task is significantly better (similar results also hold for Llama-2-7B in Appendix B.3). This suggests that pretrained real-world LLMs may also struggle to learn new input-output mappings from context; instead, ICL appears to be more adept at retrieval tasks. To show that the failure in the synthetic vector classification task is mainly due to its OOD nature instead of some other factors that make it difficult to learn, we train a GPT-2 to perform the same task in Appendix B.2 and find that the task can be well addressed after training.

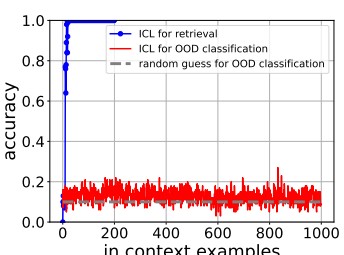

Figure 6: The ICL accuracy of Llama-3-8B on the synthetic tasks.

## 5 THE ALGORITHM SELECTION MECHANISM EXISTS BROADLY WHEN EVALUATED ON OOD TASKS

Real-world LLMs are pretrained on a huge corpus that could contain massive tasks. Bai et al. (2023); Yadlowsky et al. (2023) have empirically found that the ICL performance of Transformers trained on multiple tasks approaches the optimal pretraining function when evaluated on one of the training tasks. In this section, we will show that this algorithm-selection phenomenon of ICL broadly persists when evaluated on OOD tasks, regardless of the test distribution, and provide a comprehensive theoretical characterization of the algorithm-selection mechanism.

**The Model pretrained on a single task vs. the model pretrained on multiple tasks.** In Figure 7, we compare the performance of GPT-2 models trained on a single task—linear regression (LR), quadratic regression (QR), 2-layer ReLU neural network (ReLU NN) regression—against the model trained on all three tasks when encountering four kinds of OOD tasks. We also plot the error of a 2-layer ReLU NN trained by GD (dashed blue line).

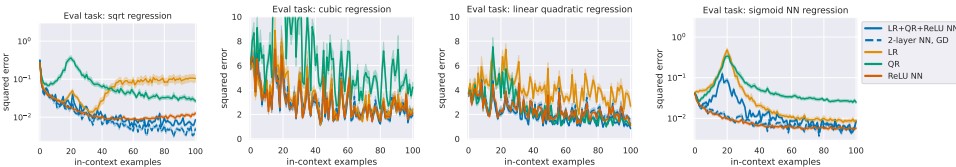

(a) Evaluated on SRR  (b) Evaluated on CR  (c) Evaluated on LQR  (d) Evaluated on Sigmoid NN

Figure 7: The ICL performance of models trained on the individual task: linear regression (LR), quadratic regression (QR), 2-layer ReLU network (ReLU NN) regression, and the model trained on the mixture of the three tasks (LR+QR+ReLU NN). The evaluation functions are (a) square root regression (SRR), (b) cubic regression (CR), (c) linear+quadratic regression (LQR), and (d) 2-layer Sigmoid network (Sigmoid NN). The details of these evaluation tasks are in Appendix C.1). The performance of the model trained on the mixed tasks is close to that of the model trained on the single task that performs the best on the evaluation task.

**Observations.** 1) The ICL performance of the model trained on mixed tasks (LR+QR+ReLU NN) is close to the performance of the model trained on a single task with the lowest test error on the evaluation task. This suggests that ICL can automatically select the best pretraining functions according to the downstream context. 2) The ICL performance of training on the ReLU NN function class aligns well with the ReLU NN model trained by GD, demonstrating that our findings in Section 3.1 still hold when the Transformer is trained on a mixture of multiple tasks.

## 5.1 THEORETICALLY REVEALING THE MECHANISM OF ALGORITHM SELECTION

In this section, we will provide theoretical insights into the underlying mechanism of the algorithm selection of ICL. We find there simultaneously exist two parallel mechanisms: the **Low-test-error preference** and the **Similar-input-distribution preference.**

**A mixed Gaussian pretraining dataset of multiple tasks.** We adopt the theoretical framework by Lin & Lee (2024). Consider a noisy linear regression pretraining dataset with the inputs and task weights following the mixed Gaussian distribution:

**Assumption 5.1.** (Mixed Gaussian pretraining data) "A pretraining task" corresponds to a component of the mixed Gaussian distribution containing a mean of the input mean $\boldsymbol{\mu}_m$ and a task weight mean $\boldsymbol{w}_m$. The input means $\boldsymbol{\mu}$ and task weights $\boldsymbol{w}$ are sampled from a mixed Gaussian distribution: $P(\boldsymbol{\mu}, \boldsymbol{w}) = \sum_{m=1}^{M} \pi_m \mathcal{N}(\boldsymbol{\mu}|\boldsymbol{\mu}_m, \sigma_\mu^2 \boldsymbol{I}) \cdot \mathcal{N}(\boldsymbol{w}|\boldsymbol{w}_m, \sigma_w^2 \boldsymbol{I})$, where $\sum_{m=1}^{M} \pi_m = 1$, $0 < \pi_m < 1$ and $\|\boldsymbol{\mu}_m\| = \|\boldsymbol{w}_m\| = 1, \forall m$. The process of sampling a training sequence $\mathcal{S}_T = [\boldsymbol{x}_1 \oplus y_1 \oplus ... \oplus \boldsymbol{x}_T \oplus y_T]$ $\boldsymbol{x}_i$ and $y_i$ is as follows: 1) Sample the input mean $\boldsymbol{\mu}$ and the task weight $\boldsymbol{w}$ according to $P(\boldsymbol{\mu}, \boldsymbol{w})$. 2) Sample $\boldsymbol{x}_i$ from $P(\boldsymbol{x}|\boldsymbol{\mu}) = \mathcal{N}(\boldsymbol{x}|\boldsymbol{\mu}, \sigma_x^2 \boldsymbol{I})$. 3) Sample $y_i$ from $P(y|\boldsymbol{x}, \boldsymbol{w}) = \mathcal{N}\left(y|\langle \boldsymbol{x}, \boldsymbol{w}\rangle, \sigma_y^2\right)$. Define $\delta_\mu = \frac{\sigma_\mu^2}{\sigma_x^2}$ and $\delta_w = \frac{\sigma_w^2}{\sigma_y^2}$. Denote this pretraining distribution as $P_{tr}$.

According to the Corollary 2 of Lin & Lee (2024) (see Lemma E.1 in Appendix E), the closed-form prediction of the model trained on the pretraining data of Assumption 5.1, given the testing context and the query $\boldsymbol{x}_{T+1}$, remains a Gaussian mixture of the reweighted pretraining task weights: $\mathcal{F}^* := \left\langle \boldsymbol{x}_{T+1}, \sum_{m=1}^{M} \tilde{\pi}_m \tilde{\boldsymbol{w}}_m \right\rangle$, where $\tilde{\pi}_m$ and $\tilde{\boldsymbol{w}}_m$ are the posterior variables of $\pi_m$ and $\boldsymbol{w}_m$ given the downstream context. Hence, to analyze how ICL selects pretraining priors, the key lies in uncovering how $\tilde{\pi}_m$ evolves after seeing the test context. First, we introduce Lemma 5.2 from Lin & Lee (2024) that characterizes the ratio of the reweighted weight of two pretraining tasks:

**Lemma 5.2.** *(Appendix H.1 of Lin & Lee (2024)) Consider any two different pretraining component $\alpha$ and $\beta$, given a test context $\mathcal{S}_T \oplus \boldsymbol{x}_{T+1}$ and the well-pretrained model $M^*$, the ratio between the*

*weights of the two task priors $\tilde{\pi}_\alpha/\tilde{\pi}_\beta$ in $M^*$'s ICL prediction can be decomposed into two terms: $\tilde{\pi}_\alpha/\tilde{\pi}_\beta = \frac{\pi_\alpha}{\pi_\beta} \exp(\Psi_{\boldsymbol{\mu}}(\alpha, \beta) + \Psi_{\boldsymbol{w}}(\alpha, \beta))$, where*

$$\Psi_{\boldsymbol{\mu}}(\alpha, \beta) = \left( \sum_{i=1}^{T+1} \|\boldsymbol{\mu}_\beta - \boldsymbol{x}_i\|^2 - \sum_{i=1}^{T+1} \|\boldsymbol{\mu}_\alpha - \boldsymbol{x}_i\|^2 \right) / \left( 2\sigma_x^2 \left( 1 + (T+1)\delta_\mu \right) \right). \quad (1)$$

*((ICL favors the pretraining function with similar input distribution to the test data) Further, assuming the test in-context examples $\boldsymbol{x}_i \sim \mathcal{N}(\boldsymbol{\mu}^*, \tau_x^2 \boldsymbol{I})$, if $\|\boldsymbol{\mu}_\beta - \boldsymbol{\mu}^*\|^2 - \|\boldsymbol{\mu}_\alpha - \boldsymbol{\mu}^*\|^2 \geq 0$ holds, then as the context length $T \to \infty$, the first term $\Psi_{\boldsymbol{\mu}}(\alpha, \beta) \to (\|\boldsymbol{\mu}_\beta - \boldsymbol{\mu}^*\|^2 - \|\boldsymbol{\mu}_\alpha - \boldsymbol{\mu}^*\|^2)/2\sigma_\mu^2 \geq 0.$*

However, Lin & Lee (2024) did not analyze how the second term $\Psi_{\boldsymbol{w}}(\alpha, \beta)$ would evolve given any downstream task, which we will demonstrate to play an important role in the algorithm selection mechanism. In the following theorem, we prove that $\Psi_{\boldsymbol{w}}(\alpha, \beta)$ converges to a non-negative value when the pretraining function class $\alpha$ performs better on the downstream context than $\beta$.

**Theorem 5.3.** *(ICL favors the pretraining function with low error on the context, proof is in Appendix E.3) Given the context $\boldsymbol{S}_T$, if the empirical risk of implementing a function of the pretraining task $\alpha$ is less than that of $\beta$, i.e., $\frac{1}{T} \sum_{i=1}^{T} \|\boldsymbol{w}_\beta \boldsymbol{x}_i - y_i\|^2 - \|\boldsymbol{w}_\alpha \boldsymbol{x}_i - y_i\|^2 \geq 0$, then, under some mild Assumptions E.2 on the distribution of $\boldsymbol{S}_T$, we have $\Psi_{\boldsymbol{w}}(\alpha, \beta) \geq 0$.*

*Combining Lemma 5.2, if the downstream inputs $\boldsymbol{x}_i$, $\boldsymbol{x}_i \sim \mathcal{N}(\boldsymbol{\mu}^*, \tau_x^2 \boldsymbol{I})$ and $\|\boldsymbol{\mu}_\beta - \boldsymbol{\mu}^*\|^2 - \|\boldsymbol{\mu}_\alpha - \boldsymbol{\mu}^*\|^2 \geq 0$ hold, then as $T \to \infty$, we have $\tilde{\pi}_\alpha/\tilde{\pi}_\beta \geq \pi_\alpha/\pi_\beta$.*

**Summary of the algorithm-selection mechanism.** Lemma 5.2 and Theorem 5.3 together elucidate the algorithm-selection mechanism of ICL. According to Lemma E.1, the ICL prediction of the model pretrained on the mixed Gaussian data will be a reweighted combination of the pretraining task vectors $\boldsymbol{w}_i$. Whether the ratio between the weights of two pretraining tasks, $\tilde{\pi}_\alpha/\tilde{\pi}_\beta$, given a downstream context, exceeds the original ratio $\pi_\alpha/\pi_\beta$ depends on two factors: 1) whether the pretraining input distribution of $\alpha$ is closer to the downstream input distribution than that of $\beta$; 2) whether the task function of $\alpha$ induces lower test error in downstream context than that of $\beta$. When both conditions are met, we have $\tilde{\pi}_\alpha/\tilde{\pi}_\beta \geq \pi_\alpha/\pi_\beta$, indicating that ICL prefers $\alpha$ over $\beta$ in its predictions. We leave the discussions of the advantage of our theory result in Appendix A.5 and offer an intuitive Bayesian interpretation of the algorithm selection in Appendix A.4.

## 5.2 Empirical Validation of the Algorithm-selection Mechanism of ICL

Now we validate our theoretical findings regarding ICL's algorithm-selection mechanism in OOD tasks by conducting numerical experiments following Lin & Lee (2024). In Figure 8a and 8b, the training data is a linear regression Gaussian mixture with four components (see Assumption 5.1), while the test function is a two-layer ReLU network (Appendix C.1). Both the training and the test data are in ICL format. We plot the test error of using each pretraining task function to predict the downstream function (the first row of Figure 8a and 8b), the weights for each pretraining function during ICL inference (the second row), and the test error of the pretrained ICL model with the closed form prediction derived in Lemma E.1 (the third row). We evaluate five different noise levels ($\delta_x = \delta_w \in \{1/81, 1/9, 1, 9, 81\}$, greater value means larger noise) and consider two settings described below that respectively validate the two mechanism in Section 5.1.

**Low-test-error preference of ICL.** To validate Theorem 5.3, we ensure that the distributional distances between the inputs of each training task and the test data remain consistent. Specifically, all $\boldsymbol{x}_i$ in both training and test data are sampled from $\mathcal{N}([0, 0, 0]^\top, \sigma_x^2 \boldsymbol{I})$. The task weights for different pretraining tasks vary, as detailed in the top half of Table 1. In this setup, only the test error of the pretraining functions influences algorithm selection. From the top two rows of Figure 8a, we can observe a clear negative correlation between the task weight and the test error of the corresponding pretraining task. This result supports Theorem 5.3, confirming that ICL prefers the pretraining functions with a low test error. Also, it's consistent with the observations in Figure 7.

**Similar-input-distribution preference of ICL.** We also empirically validate Lemma 5.2 in Figure 8b. In this case, the distributional distances between the input of different pretraining tasks and that of the test context vary: the distances of different tasks are ordered from largest to smallest as $1 > 2 > 3 > 4$, while the test errors of different pretraining functions are almost the same (detailed setup is in the bottom half of Table 1). As shown in the bottom two rows in Figure 8b, the task

weight $\tilde{\pi}_i$ is positively correlated with the similarity between the training and test input distribution. This is consistent with Lemma 5.2 which demonstrates that ICL prefers to select the pretraining function whose input distribution is close to the downstream one.

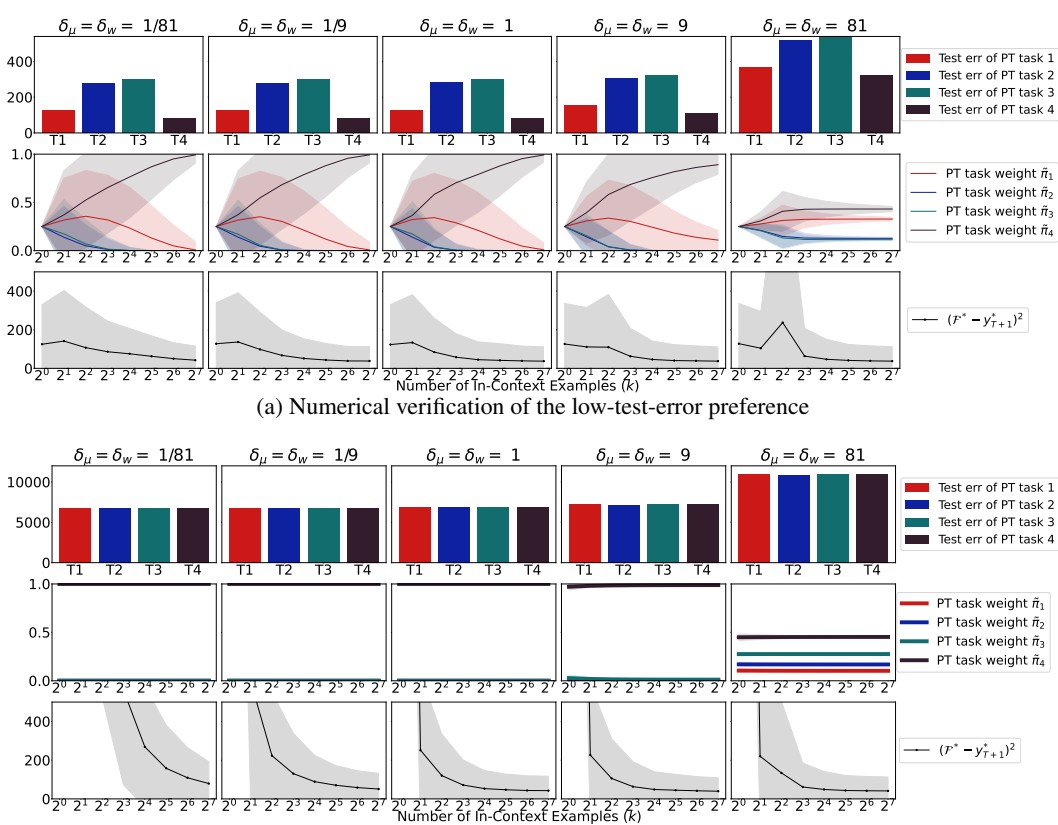

(a) Numerical verification of the low-test-error preference

(b) Numerical verification of the similar-input-distribution preference

Figure 8: Empirical validation of the algorithm-selection mechanism of ICL. The first rows: the test error of the four pretraining functions. The mid rows: the weights of each pretraining function in the closed-form downstream ICL prediction (given by Lemma E.1). The last rows: the test error of the pretrained ICL model with the closed form prediction derived in Lemma E.1. **Observations.** 1) In the first two rows of Figure 8a, the value of the task weight $\tilde{\pi}_i$ is negatively correlated with the test error of pretraining task $i$. 2) In the first two rows of Figure 8b the task weights are negatively correlated with the distance between the training and testing input distribution.

Table 1: Experiment setting of Figure 8a and Figure 8b. "PT" and "DS" are short for "pretraining" and "downstream", respectively.

| Experiment | DS inputs | PT task id | PT input distribution | PT task functions | PT-DS input distance |
|---|---|---|---|---|---|
| Figure 8a | $\mathcal{N}([0,0,0]^\top, \sigma_x^2 \boldsymbol{I})$ | 1 | $\mathcal{N}([0,0,0]^\top, \sigma_x^2 \boldsymbol{I})$ | $\mathcal{N}([5,5,5]^\top, \sigma_w^2 \boldsymbol{I})$ | 0 |
| | | 2 | $\mathcal{N}([0,0,0]^\top, \sigma_x^2 \boldsymbol{I})$ | $\mathcal{N}([-5,5,5]^\top, \sigma_w^2 \boldsymbol{I})$ | 0 |
| | | 3 | $\mathcal{N}([0,0,0]^\top, \sigma_x^2 \boldsymbol{I})$ | $\mathcal{N}([-5,5,-5]^\top, \sigma_w^2 \boldsymbol{I})$ | 0 |
| | | 4 | $\mathcal{N}([0,0,0]^\top, \sigma_x^2 \boldsymbol{I})$ | $\mathcal{N}([-5,-5,-5]^\top, \sigma_w^2 \boldsymbol{I})$ | 0 |
| Figure 8b | $\mathcal{N}([-4,-4,-4]^\top, \sigma_x^2 \boldsymbol{I})$ | 1 | $\mathcal{N}([5,5,5]^\top, \sigma_w^2 \boldsymbol{I})$ | $\mathcal{N}([1,1,1]^\top, \sigma_w^2 \boldsymbol{I})$ | 15.59 |
| | | 2 | $\mathcal{N}([-5,5,5]^\top, \sigma_w^2 \boldsymbol{I})$ | $\mathcal{N}([1,1,1]^\top, \sigma_w^2 \boldsymbol{I})$ | 12.77 |
| | | 3 | $\mathcal{N}([-5,5,-5]^\top, \sigma_w^2 \boldsymbol{I})$ | $\mathcal{N}([1,1,1]^\top, \sigma_w^2 \boldsymbol{I})$ | 9.11 |
| | | 4 | $\mathcal{N}([-5,-5,-5]^\top, \sigma_w^2 \boldsymbol{I})$ | $\mathcal{N}([1,1,1]^\top, \sigma_w^2 \boldsymbol{I})$ | 1.73 |

### 5.3 VERIFYING THE ALGORITHM-SELECTION MECHANISM ON REAL-WORLD LLMS

In this section, we investigate whether real-world LLMs can perform algorithm selection through ICL. To achieve this, we design an ambiguous sentence classification task, in which each sentence

can be classified based on one of three aspects: "sentiment", "type", or "location". For each ICL sequence, we select one of the aspects as the classification criterion and map the label words to meaningless strings. For instance, if we choose to classify each sentence according to its sentiment, then "positive", "neutral", and "negative" are mapped to "RqF", "IwZ", and "SdK", respectively. Detailed experimental setups are in Appendix C.5. We compute the top-5 accuracy of different classification criteria. The results in Figure 9 show that as the context length increases, the LLM finds the most appropriate criterion, exhibiting the low-test-error preference.

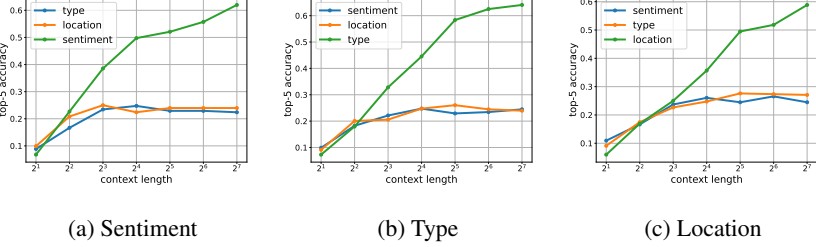

(a) Sentiment          (b) Type          (c) Location

Figure 9: The top-5 accuracy of using (a)"sentiment", (b)"type", or (c)"location" as the classification criterion for in-context examples in a test prompt. The accuracy of using the true underlying criterion to predict is significantly higher than the other two. This suggests that LLMs can perform algorithm selection in natural language tasks.

---

**Summary of the Empirical Results & Connections with the Existing Theories (III)**

Despite the impressive empirical performance of real-world LLMs in solving seemingly novel tasks through ICL, we observe that when faced with an OOD task, ICL operates by identifying the most suitable pretraining meta-distribution based on test error and input distribution discrepancies, and then attempts to find an optimal solution within that meta-distribution. Notably, this process occurs consistently, independent of the downstream test distribution. See Appendix A.4 for the potential connection between such empirical observations and the Bayesian framework work.

---

## 6 CONCLUSION

In this work, we empirically find that Transformers struggle to generalize beyond the pretraining function classes when given downstream in-context examples of OOD tasks. Instead, ICL tries to seek a near-optimal solution within the pretraining function classes. We further investigate the widely observed capability of ICL to perform classification. We reveal that it is a composition of ID prediction and retrieval rather than an OOD generalization ability. We also examine ICL's performance on OOD tasks after pretraining on multiple tasks. Our theoretical and empirical analysis reveals ICL's preference for low-test-error functions, i.e., ICL tends to implement pretraining function classes with low test error in the test context. This finding highlights two key factors that determine how ICL will implement the prediction function based on the testing context and pretraining tasks: the distance between the training and testing input distributions, and the ability of a pretraining function to solve the test task.

## ACKNOWLEDGMENT

Yisen Wang was supported by National Key R&D Program of China (2022ZD0160300), National Natural Science Foundation of China (92370129, 62376010), and Beijing Nova Program (20230484344, 20240484642). Xianghua Ying was supported by the National Natural Science Foundation of China (NSFC) under Grant No. 62371009, and Beijing Natural Science Foundation under Grant No. L247029. Yifei Wang was supported in part by the NSF AI Institute TILOS, and an Alexander von Humboldt Professorship.

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

# A   COMPARISON WITH RELATED WORKS AND ADDITIONAL DISCUSSIONS

## A.1   THE CAPABILITY OF ICL TO LEARN NEW TASKS

Besides studies indicating that ICL can learn new weights of linear regression (Garg et al., 2022; Raventós et al., 2023; Zhang et al., 2023a; Akyürek et al., 2023), other research has found that LLMs can tackle tasks that are unlikely to have been encountered during pretraining. For example, Pan (2023) showed that LLMs perform better than random guessing on classification tasks with meaningless labels. Kossen et al. (2024) demonstrate that ICL can identify authorship based on writing style in private communication messages not included in the pretraining corpus. Additionally, Vacareanu et al. (2024) found that large-scale LLMs can learn various linear and non-linear functions from context. We argue that these findings do not contradict our work. While the LLMs may not have seen exactly the same tasks, there is no guarantee that they haven't encountered tasks from a similar distribution in their pretraining corpus. For instance, the LLMs could have been pretrained on a corpus containing authorship identification tasks or on statistical data encompassing different functions. Our work does not claim that ICL cannot generalize to new task instances of a seen distribution; rather, it highlights the limitation in generalizing to an unseen input-label distribution. Additionally, Yadlowsky et al. (2023) finds that ICL struggles to generalize to testing function classes that are unseen during training (e.g., convex combinations or extreme versions of the pretraining functions). They didn't delve into how ICL behaves on OOD data, while we reveal that it implements the pretraining functions.

## A.2   THE ALGORITHM-SELECTION MECHANISM OF ICL

Recent works by Bai et al. (2023); Wang et al. (2024b) have uncovered the algorithm selection phenomenon, demonstrating that Transformers pretrained on both linear regression and classification tasks perform well when presented with the context of either task during ICL inference. Theoretically, they show that a Transformer with specific parameters can achieve algorithm selection. Yadlowsky et al. (2023) empirically found that ICL selects the optimal pretraining function class after observing in-context examples from a function class present in the pretraining data mixture. However, the algorithm selection experiments in these studies are limited to scenarios where *the test functions are among the training functions*. In this work, we empirically and theoretically demonstrate that the algorithm selection phenomenon broadly occurs when given downstream context from arbitrary function classes. To the best of our knowledge, we are the first to reveal the factors that determine the selection process.

## A.3   THE BAYESIAN-OPTIMAL PERSPECTIVE FOR UNDERSTANDING ICL

Many studies have found that ICL makes Bayes-optimal predictions (Xie et al., 2022; Wies et al., 2024; Zhang et al., 2023b; Lin & Lee, 2024). However, these works have certain limitations that may reduce their practical applicability in predicting ICL behavior in general scenarios. 1) Limited empirical verification. Wies et al. (2024) and Zhang et al. (2023b) lack empirical verification of their theory on real deep Transformer models; 2) Limited theoretical settings: in-distribution tasks. Wies et al. (2024) assumes the downstream tasks are components of the pretraining distribution; Xie et al. (2022) assumes that the latent concept of the test task $\theta^*$ is within the pretraining task set $\Theta$; In Lin & Lee (2024), the training and testing tasks are all linear regression with weights sampled from Gaussian distribution. 3) Limited implications of the theoretical results: although Xie et al. (2022); Zhang et al. (2023b) prove that ICL can infer a task concept $\theta$ based on the downstream test context $S_{test}$, they don't reveal how $S_{test}$ concretely affects the posterior distribution $P(\theta|S_{test})$ of the latent task concept $\theta$ inferred by the model that determines the downstream ICL prediction, especially when the true downstream task $\theta^*$ is OOD. Our work verifies and extends previous findings to a more general setting by using real deep Transformers and evaluating ICL on OOD tasks that significantly differ from the training tasks. For the first time, we also reveal how the interaction between the downstream distribution and the pretraining distribution affects ICL predictions (see Section 5).

In contrast, Raventós et al. (2023) claim that ICL can exhibit non-Bayesian properties. They empirically demonstrate that when given sufficiently diverse pretraining tasks (linear regression vectors), ICL can outperform the Bayesian estimator on a new test distribution. However, the distributional shift in their setup might not be substantial enough to show that ICL can truly adapt to a new down-

stream distribution, which is considered to be "non-Bayesian" by Raventós et al. (2023). In their setting, both the test and training vectors are sampled from the standard Gaussian distribution, and the only source of "distributional shift" comes from the finite size of the training set, which can only partially reflect the test distribution. Our work refutes their findings by showing that when the test distribution is significantly shifted, increasing the number of ID tasks may not help ICL generalize to it.

### A.4    The Bayesian Interpretation for Our Empirical Findings

Although current Bayesian theories for ICL are too vacuous to predict the performance of deep Transformers on real OOD tasks (see Section A.3), the Bayesian framework shows promise as a potential lens for interpreting our empirical findings. Here we provide some intuitive interpretations for the findings in Section 3, 4, and 5 from a Bayesian perspective.

Consider the predicted distribution $p_\theta(\boldsymbol{y}_T|\boldsymbol{x}_{1:T})$ given by a pretrained model $\theta$. If we assume that ICL makes Bayesian-like predictions over the test context as Xie et al. (2022); Wies et al. (2024); Zhang et al. (2023b); Lin & Lee (2024) suggested, then the model will first infer a task concept $\phi$ based on the given context $\boldsymbol{x}_{1:T-1}$ and predict $\boldsymbol{y}_T$ using $\phi$ and $\boldsymbol{x}_{1:T}$, i.e.,

$$p_\theta(\boldsymbol{y}|\boldsymbol{x}_{1:T}) = p_\theta(\boldsymbol{y}|\boldsymbol{x}_{1:T}, \phi)p_\theta(\phi|\boldsymbol{x}_{1:T-1}) \tag{2}$$

To explain the results in Section 3 and Section 5, since the true downstream task $\phi^*$ is unseen during pretraining, the inferred posterior distribution $p_\theta(\phi|\boldsymbol{x}_{1:T-1})$ assigns probability mass only to tasks $\phi$ within the pretraining distribution that maximize $p_\theta(\boldsymbol{y}|\boldsymbol{x}_{1:T})$. This accounts for why ICL can only make in-distribution predictions, as shown in Figure 1 in Section 3, and why ICL prefers pretraining priors with low test error and input distributions similar to those in the test context (Section 5). Once a task $\phi$ seen during pretraining is identified as best fitting the test context $\boldsymbol{x}_{1:T-1}$, the model refines its predictions based on this context (in Figure 1 and Figure 7, the test error decreases as the number of in-context examples increase). This refinement corresponds to the factor $p_\theta(\boldsymbol{y}|\boldsymbol{x}_{1:T}, \phi)$, explaining how ICL optimizes predictions within its pretraining distribution.

In Section 4, the underlying task concept $\phi$ acts as a similarity metric that allows the model to retrieve examples from the context that align with the query. Training on more abstract labels improves the model's ability to estimate a more accurate $\phi$, which explains the results in 3 and Figure 4. When the test task is ID, even with OOD labels, ICL can succeed by leveraging the learned $\phi$ to predict the true label. It accomplishes this by retrieving an example $\boldsymbol{x}_i$ from the context that is similar to the query under the $\phi$ metric. However, when the underlying task $\phi^*$ is OOD, the model fails because the learned similarity metric no longer applies effectively (Figure 5).

### A.5    Discussion of the Setup of Our Theory

Notably, our theoretical result in Section 5.1 applies to any model architecture, while previous theoretical works of understanding ICL often adopt Transformers with oversimplified assumptions on their parameters or structures (Ahn et al., 2023; Zhang et al., 2023a; Huang et al., 2023; Collins et al., 2024). Additionally, our analysis shows that models pretrained on the ICL tasks can implement algorithm selection during ICL inference following Lin & Lee (2024). In contrast, prior work on algorithm selection (Bai et al., 2023) only shows that a specific set of parameters in a simplified ReLU Transformer can enable algorithm selection. However, the parameter construction is complex and somewhat tricky, and there is no theoretical or experimental guarantee that Transformers exhibiting algorithm selection will necessarily implement these parameters.

## B    Additional Experimental Results

### B.1    Understanding the Effect of Training on More Diverse Retrieval Tasks from the Attention Scores

To further validate that the retrieval ability is evoked after trained on more random mappings, following Crosbie & Shutova (2024), we construct another retrieval task and visualize the *prefix matching score* of all attention heads of the three pretrained models in Figure 10. The prefix matching score

is calculated by averaging the attention values from each token $t_i$ to the tokens after the same token as $t_i$ in earlier positions in the sequence, which correlates positively with the retrieval performance (Singh et al., 2024b). In Figure 10, we observe that the model best at the retrieval task in Figure 3 exhibits more heads with high matching scores (also known as "induction heads" (Song et al., 2024; Singh et al., 2024a; Ren et al., 2024; Crosbie & Shutova, 2024; Edelman et al., 2024)), further demonstrating it gains the retrieval ability by training on more retrieval sequences.

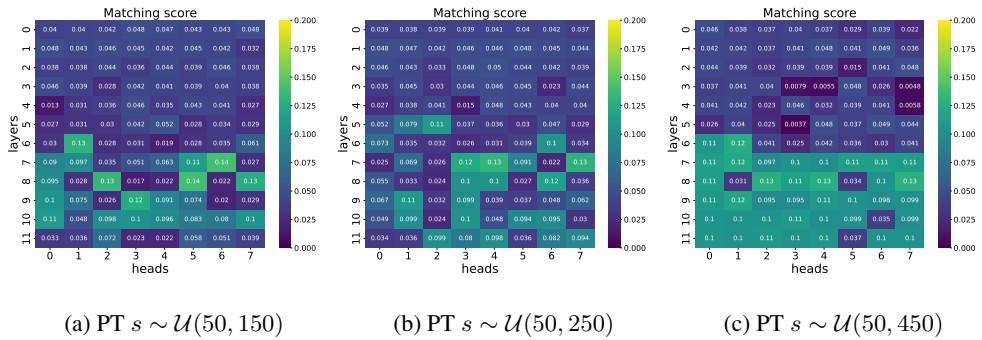

(a) PT $s \sim \mathcal{U}(50, 150)$      (b) PT $s \sim \mathcal{U}(50, 250)$      (c) PT $s \sim \mathcal{U}(50, 450)$

Figure 10: The matching score of all attention heads of models trained on the retrieval task. "PT" denotes "pretrained on". Each subfigure corresponds to a different pretrained model. The model of (c) exhibits more heads with high matching scores, which is also the most performant model in the retrieval task in Figure 3.

### B.2 THE SYNTHETIC VECTOR CLASSIFICATION IS NOT THAT HARD TO SOLVE IF IT'S IN DISTRIBUTION

To show the failure in the synthetic vector classification in Section 4.3 is mainly due to its OOD nature rather than it's intrinsically too hard to learn, we train a GPT-2 to perform the same task as in Section 4.3. In this task, the $x_i$ and $y_i$ are generated in the same way as Section 4.3. The only modification is that we use a smaller predefined vector embedding $E' \in \mathbb{R}^{10000 \times 20}$ ($E_{llama} \in \mathbb{R}^{32000 \times 4096}$ in the experiment in Section 4.3). The results in Figure 11 show that when $W$ has been encountered during pretraining, ICL can well address this task.

### B.3 EVALUATING THE SYNTHETIC OOD CLASSIFICATION TASK ON LLAMA-2-7B

We also evaluate Llama-2-7B on the same OOD vector classification task and the retrieval task as in Section 4.3. Figure 12 shows the same observations as in Figure 6 that the LLM can well address the retrieval task but fails to learn the OOD function $W$. In this experiment, we set the length of the repeating sequence to be 10. We can observe that the accuracy of retrieval rapidly increases after seeing 10 in-context examples. This demonstrates that learning novel functions from the context is challenging for real-world pretrained LLMs, but the LLMs are good at retrieving.

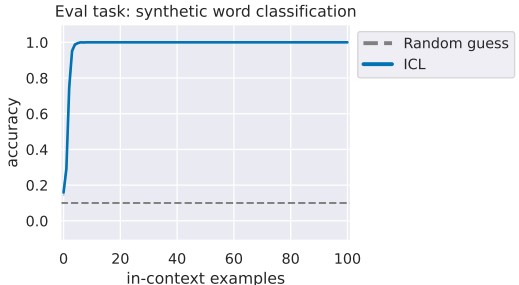

Figure 11: Test error of the GPT-2 trained and evaluated on the same synthetic OOD vector classification task as in Section 4.3.

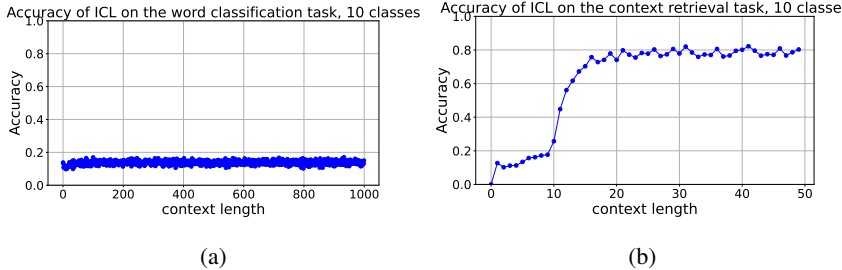

Figure 12: The ICL accuracy of Llama-2-7B on the synthetic tasks. (a) the synthetic vector classification task. (b) the synthetic word retrieval task.

### B.4 WILL GENERALIZATION CAPABILITIES EMERGE FROM INCREASING THE NUMBER OF TRAINING TASKS?

Recent work by Raventós et al. (2023) empirically demonstrates that when both the training and test tasks are linear regression, and the number of training vectors exceeds a certain "task diversity threshold" (approximately $2^{14} \sim 2^{15}$), ICL can generalize from a finite training set sampled biasedly from $\mathcal{N}(0,1)$ to the test distribution $P_{\text{test}} = \mathcal{N}(0,1)$ (see Appendix A.3 for details). We investigate whether similar phenomena persist for test tasks with larger distributional shifts. We train models using varying numbers of linear regression vectors and evaluate them on quadratic and ReLU neural network regression tasks. In Figure 13, we find that training on a vast number of ID functions does not yield any improvements, providing further evidence that ICL may struggle to achieve OOD generalization.

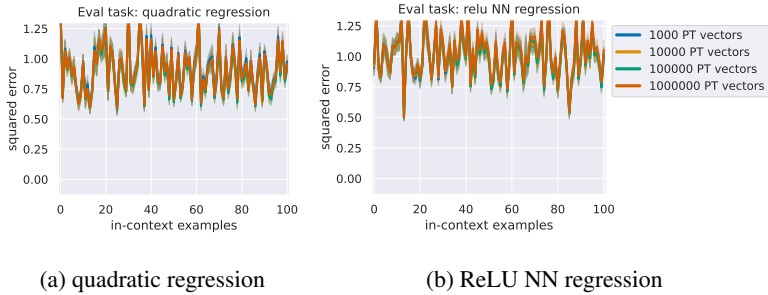

(a) quadratic regression        (b) ReLU NN regression

Figure 13: The ICL test error of models trained on different numbers of linear regression vectors. Even if the number of training vectors (up to $1,000,000 \approx 2^{20}$) surpasses the threshold ($2^{14} \sim 2^{15}$) reported by Raventós et al. (2023), no model exhibits generalization to OOD function classes.

## C EXPERIMENTAL DETAILS

### C.1 EXPERIMENTAL DETAILS IN SECTION 3.1 AND SECTION 5

**Definitions of the function classes.** The function classes in Figure 1 and Figure 7 are:

- Linear regression: $y_i = \boldsymbol{w}^\top \boldsymbol{x}_i$, where $\boldsymbol{w}, \boldsymbol{x}_i \in \mathbb{R}^d$ and $\boldsymbol{w}, \boldsymbol{x}_i \sim \mathcal{N}(0,1)$.
- Quadratic regression: $y_i = \boldsymbol{w}^\top (\boldsymbol{x}_i)^2$, where $\boldsymbol{w}, \boldsymbol{x}_i \in \mathbb{R}^d$ and $\boldsymbol{w}, \boldsymbol{x}_i \sim \mathcal{N}(0,1)$, $(\boldsymbol{x}_i)^2$ denotes the element-wise square of $\boldsymbol{x}_i$.
- 2-layer ReLU network regression: $y_i = \boldsymbol{w}_1^\top \text{ReLU}(\boldsymbol{w}_2 \boldsymbol{x}_i)$, where $\boldsymbol{w}_1 \in \mathbb{R}^{d'}$, $\boldsymbol{w}_2 \in \mathbb{R}^{d' \times d}$, and $\boldsymbol{x}_i \in \mathbb{R}^d$. $\boldsymbol{w}_1, \boldsymbol{w}_2, \boldsymbol{x}_i \sim \mathcal{N}(0,1)$.
- Square root linear regression: $y_i = \boldsymbol{w}^\top \sqrt{\boldsymbol{x}_i}$, where $\boldsymbol{w}, \boldsymbol{x}_i \in \mathbb{R}^d$ and $\boldsymbol{w}, \boldsymbol{x}_i \sim \mathcal{N}(0,1)$, $(\boldsymbol{x}_i)^2$ denotes the element-wise square root of $\boldsymbol{x}_i$.
- Cubic linear regression: $y_i = \boldsymbol{w}^\top (\boldsymbol{x}_i)^3$, where $\boldsymbol{w}, \boldsymbol{x}_i \in \mathbb{R}^d$ and $\boldsymbol{w}, \boldsymbol{x}_i \sim \mathcal{N}(0,1)$, $(\boldsymbol{x}_i)^2$ denotes the element-wise cube of $\boldsymbol{x}_i$.

- Linear+quadratic regression: $y_i = \boldsymbol{w}_1^\top (\boldsymbol{x}_i)^2 + \boldsymbol{w}_2^\top \boldsymbol{x}_i$, where $\boldsymbol{w}_1$, $\boldsymbol{w}_2$, $\boldsymbol{x}_i \in \mathbb{R}^d$ and $\boldsymbol{w}_1$, $\boldsymbol{w}_2$, $\boldsymbol{x}_i \sim \mathcal{N}(0, 1)$.

- 2-layer Sigmoid network: $y_i = \boldsymbol{w}_1^\top \text{Sigmoid}(w_2 \boldsymbol{x}_i)$, where $\boldsymbol{w}_1 \in \mathbb{R}^{d'}$, $\boldsymbol{w}_2 \in \mathbb{R}^{d' \times d}$, and $\boldsymbol{x}_i \in \mathbb{R}^d$. $\boldsymbol{w}_1$, $\boldsymbol{w}_2$, $\boldsymbol{x}_i \sim \mathcal{N}(0, 1)$.

**Baseline models in Figure 1.** The models of each pretraining hypothesis class are implemented by training a neural network that yields functions of that hypothesis class. For example, a linear regression weight $w$ can be implemented by a single linear layer. The models are optimized using SGD with learning rate 1e-3 for 1000 steps.

## C.2 EXPERIMENTAL DETAILS FOR SECTION 3.2

For the reversed-label experiment, we choose four tasks: Antonym, Capital-country, English-French, and English-German. The original datasets are adopted from Todd et al. (2024). The top-1 accuracy is computed as follows: compute the top-1 accuracy for each token predicted by the model, based on the token length of the ground-truth label word. For each context length, we compute the average accuracy over 128 test examples.

## C.3 EXPERIMENTAL DETAILS FOR SECTION 4.1

We now provide additional details regarding the experiments of Figure 10 . Following Crosbie & Shutova (2024), we generated a dataset consisting of 100 sequences of random tokens, each containing repeated sub-sequences. The task is to predict the next token that follows the last token in each sequence. This task can only be completed by retrieving the last token from the context and predicting its subsequent token.

## C.4 EXPERIMENTAL DETAILS FOR SECTION 4.3

We uniformly sample 1000 word vectors $\boldsymbol{x}_i \in \mathbb{R}^d$ from the word embedding $E \in \mathbb{R}^{N \times d}$ of the pretrained Llama-3-8B, where $N = 128256$ and $d = 4096$. Then we sample a task weight $\boldsymbol{W} \in \mathbb{R}^{d' \times C}$ from standard Gaussian distribution that only takes the first $d'$ dimensions of $\boldsymbol{x}_i$ (denoted as $\boldsymbol{x}_i[: d']$) to compute a probability distribution over $C$ classes: $p_i = \boldsymbol{x}_i[: d']^\top \boldsymbol{W} \in \mathbb{R}^C$. Next, we set the label vectors $\boldsymbol{y}_i = E_{\arg\max_j p_i[j]+s} \in \mathbb{R}^d$, where $s = 10000$ is a offset. We set $d' = 30 \ll d = 4096$ to reduce the complexity of the task. Hence, $\boldsymbol{x}_i$ are classified into $C$ labels vectors $E[s : s+C]$. The predicted token of $\boldsymbol{x}_i$ is computed as: $\arg\max_j \hat{p}_i[j]$, $j \in \{s, s+1, ..., s+C-1\}$, where $\hat{p}_i$ is the output of the last linear layer of Llama-3-8B given $\boldsymbol{x}_i$.

## C.5 EXPERIMENTAL DETAILS FOR SECTION 5.3

In this section, we present some details about the setups for the ambiguous classification task. The label mapping rule is presented in Table 2. For each context length, we compute the average accuracy over 128 test examples.

Table 2: Experiment setting of Figure 8a and Figure 8b. "PT" and "DS" are short for "pretraining" and "downstream", respectively.

| Classification criterion | Original labels | Labels presented in the context |
|---|---|---|
| sentiment | "positive" "neutral" "negative" | "RqF" "IwZ" "SdK" |
| type | "science" "sports" "arts" | "RqF" "IwZ" "SdK" |
| location | "Asia" "Europe" "Africa" | "RqF" "IwZ" "SdK" |

**Prompt examples.** Here we present some in-context examples of the input prompt of using different classification criteria.

---

**Using "sentiment" as the classification criterion.**

Q: The groundbreaking discovery made by Japanese scientists has revolutionized renewable energy.
A: RqF gray*# Original label: "positive"*

Q: A chess championship occurred in Russia, featuring players from around the continent.
A: IwZ gray*# Original label: "neutral"*

---

**Using "type" as the classification criterion.**

Q: A regional basketball league was formed in Kenya to promote the sport locally.
A: IwZ gray*# Original label: "sports"*

Q: The breathtaking architectural exhibition in Dubai left visitors absolutely awestruck.
A: SdK gray*# Original label: "arts"*

---

**Using "location" as the classification criterion.**

Q: A scientific paper from Finland explores new methodologies in data analysis.
A: IwZ gray*# Original label: "Europe"*

Q: An astronomy workshop was conducted in Ethiopia for students interested in space.
A: SdK gray*# Original label: "Africa"*

---

**Accuracy computation.** For a given label, the method to calculate top-5 accuracy is as follows: compute the top-5 accuracy for each token predicted by the model, based on the token length of the ground-truth label word. For a classification criterion other than the one selected in the current sequence, to verify whether the model's prediction distribution across all test samples approaches the label distribution under that criterion, we select the permutation among all possible mappings between original labels and meaningless strings that yields the highest model prediction accuracy to compute the accuracy.

## D  EXISTING THEORETICAL EVIDENCE SUPPORTING THAT ICL MAKES ID PREDICTIONS

One recent work (Zhang et al., 2023a) theoretically proved that a one-layer linear self-attention model (LSA, defined in Appendix D) pretrained on a linear regression task will still implement the linear predictor given downstream in-context examples of arbitrary new function classes, under some assumptions on the initialization of the Transformer weight matrices. We restate the Theorem 4.2 of Zhang et al. (2023a) as Lemma D.1 below:

**Lemma D.1.** *(Theorem 4.2 of Zhang et al. (2023a), informal) Let $\mathcal{D}$ be a distribution over $(\boldsymbol{x}, y) \in \mathbb{R}^d \times \mathbb{R}$, whose marginal distribution on $x$ is $\mathcal{D}_x = \mathcal{N}(0, \Lambda)$. Assume the test prompt is of the form $P = (\boldsymbol{x}_1, y_1, \ldots, \boldsymbol{x}_T, y_T, \boldsymbol{x}_{query})$, where $(\boldsymbol{x}_i, y_i), (\boldsymbol{x}_{query}, y_{query}) \overset{i.i.d.}{\sim} \mathcal{D}$. The prediction risk on the test query $y_{query}$ of an arbitrary task satisfies:*

$$\mathbb{E}\left(\widehat{y}_{query} - y_{query}\right)^2 = \underbrace{\min_{\boldsymbol{w} \in \mathbb{R}^d} \mathbb{E}\left(\langle \boldsymbol{w}, \boldsymbol{x}_{query} \rangle - y_{query}\right)^2}_{\textit{Error of best linear predictor}} + const,$$

*where $const$ is a constant depending on the downstream context, and the expectation is over $(\boldsymbol{x}_i, y_i), (\boldsymbol{x}_{query}, y_{query}) \overset{i.i.d.}{\sim} \mathcal{D}$.*

Lemma D.1 serves as a shred of theoretical evidence that ICL can just implement the pretraining function class, while the role of the context examples is to optimize the model within the pretraining hypothesis space.

Below, we provide the necessary details of the theoretical setting of Zhang et al. (2023a).

The linear self-attention (LSA) model considered in the Theorem 4.2 of Zhang et al. (2023a) (Lemma D.1) is defined as follows:

$$f_{\text{LSA}}(E; \theta) = E + W^{PV} E \cdot \frac{E^\top W^{KQ} E}{N}, \tag{3}$$

where $E$ is the input embedding defined as follows:

$$E = E(P) = \begin{pmatrix} \boldsymbol{x}_1 & \boldsymbol{x}_2 & \cdots & \boldsymbol{x}_N & \boldsymbol{x}_{\text{query}} \\ y_1 & y_2 & \cdots & y_N & 0 \end{pmatrix} \in \mathbb{R}^{(d+1)\times(N+1)}. \tag{4}$$

$W^{PV}$ is obtained by merging the projection and value matrices into a single matrix, and $W^{KQ}$ is attained by merging the query and key matrices into a single matrix. $N$ is the context length.

Now we present the assumption on the attention weights of the linear-attention model in Lemma D.1.

**Assumption D.2.** (Assumption 3.3 in Zhang et al. (2023a), initialization). Let $\sigma > 0$ be a parameter, and let $\Theta \in \mathbb{R}^{d\times d}$ be any matrix satisfying $\left\| \Theta\Theta^\top \right\|_F = 1$ and $\Theta\Lambda \neq 0_{d\times d}$. We assume

$$W^{PV}(0) = \sigma \begin{pmatrix} 0_{d\times d} & 0_d \\ 0_d^\top & 1 \end{pmatrix}, \quad W^{KQ}(0) = \sigma \begin{pmatrix} \Theta\Theta^\top & 0_d \\ 0_d^\top & 0 \end{pmatrix}$$

The training objective is to minimize the population risk of the linear regression task:

$$L(\theta) = \lim_{B\to\infty} \widehat{L}(\theta) = \frac{1}{2}\mathbb{E}_{w_\tau, \boldsymbol{x}_{\tau,1}, \cdots, \boldsymbol{x}_{\tau,N}, \boldsymbol{x}_{\tau, \text{query}}} \left[ (\widehat{y}_{\tau, \text{query}} - \langle w_\tau, \boldsymbol{x}_{\tau, \text{query}} \rangle)^2 \right], \tag{5}$$

where $w_\tau \sim \mathcal{N}(0, I_d)$, $\boldsymbol{x}_{\tau,i}, \boldsymbol{x}_{\tau,query} \sim \mathcal{N}(0, \Lambda)$, $\widehat{y}_{\tau, \text{query}}$ is the prediction of the LSA model.

# E   THE LEMMAS, ASSUMPTION, AND PROOF FOR THEOREM 5.3

In this section, we will first present the lemmas and assumption that Theorem 5.3 depends on and the provide its proof.

## E.1   LEMMAS FOR THEOREM 5.3

The lemma below states that the closed-form prediction of the model trained on the pretraining data under Assumption 5.1, given the testing context, remains a Gaussian mixture of the reweighted pretraining task weights:

**Lemma E.1.** *(Corollary 2 of Lin & Lee (2024), closed-form ICL prediction of the pretrained model) Denote the model $M^*$ that minimizes the risk on the pretraining data of Assumption 5.1, i.e., $M^* \in \arg\min \frac{1}{T} \sum_{i=0}^{T-1} \mathbb{E}_{S_i \sim P_{tr}} \left[ \| M(\mathcal{S}_i \oplus \boldsymbol{x}_{i+1}) - y_{i+1} \|^2 \right]$, then the prediction on any sequence $\mathcal{S}_i \oplus \boldsymbol{x}_{i+1}$ by $M^*$ is as follows: $\mathcal{F}^* := M^*(\mathcal{S}_i \oplus \boldsymbol{x}_{i+1}) = \left\langle \boldsymbol{x}_{i+1}, \sum_{m=1}^{M} \tilde{\pi}_m \tilde{\boldsymbol{w}}_m \right\rangle$. where $\tilde{\pi}_m$, and $\tilde{\boldsymbol{w}}_m$ depending on both the pretraining task and the downstream context example are given in Lemma 1 of Lin & Lee (2024).*

## E.2   THE ASSUMPTION FOR THEOREM 5.3

The assumption below impose some mild requirements on the distribution of the downstream context:

**Assumption E.2.** (Assumption on the distribution of the downstream context examples.) Assume that: the minimum eigenvalue of the covariance matrix of any in-context example $\boldsymbol{x}_i$ satisfies $\lambda_{\min}(\boldsymbol{x}_i\boldsymbol{x}_i^\top) \geq 1$; $(\boldsymbol{I} + T\delta_w\boldsymbol{I})(\boldsymbol{I} + \delta_w \sum_{i=1}^{T} \boldsymbol{x}_i\boldsymbol{x}_i^\top)^{-1} = \boldsymbol{I}$; $\frac{1}{T}\sum_{i=1}^{T} 2(\boldsymbol{w}_\alpha - \boldsymbol{w}_\beta)^\top \boldsymbol{x}_i y_i \frac{1}{T}\sum_{j=1}^{T} \left( \boldsymbol{x}_j^\top \boldsymbol{x}_i \frac{y_j}{y_i} - \boldsymbol{x}_i^\top \boldsymbol{x}_i \right) \geq 0$

### E.3 PROOF FOR THEOREM 5.3

Now we restate Theorem 5.3 as the Theorem E.3 below

**Theorem E.3.** *(ICL prediction favors the pretraining function with low error on the context) Given the context $S_k$, if the empirical risk of implementing a function of the pretraining task $\alpha$ is less than that of $\beta$, i.e., $\frac{1}{T}\sum_{t=1}^{T}|\boldsymbol{w}_\beta \boldsymbol{x}_i - y_i|^2 - |\boldsymbol{w}_\alpha \boldsymbol{x}_i - y_i|^2 \geq 0$, then, under some mild Assumptions E.2, we have $\Psi_{\boldsymbol{w}}(\alpha, \beta) \geq 0$.*

*Combining Lemma 5.2, if the downstream inputs $\boldsymbol{x}_i$, $\boldsymbol{x}_i \sim \mathcal{N}(\boldsymbol{\mu}^*, \tau_x^2 \boldsymbol{I})$ and $\|\boldsymbol{\mu}_\beta - \boldsymbol{\mu}^*\|^2 - \|\boldsymbol{\mu}_\alpha - \boldsymbol{\mu}^*\|^2 \geq 0$ hold, then as $T \to \infty$, we have $\tilde{\pi}_\alpha / \tilde{\pi}_\beta \geq \pi_\alpha / \pi_\beta$.*

*Proof.* According to Lemma 1 of Lin & Lee (2024),

$$r(\alpha, \beta) = \frac{\tilde{\pi}_\alpha}{\tilde{\pi}_\beta} = \frac{\pi_\alpha C_0 c_\alpha^\mu c_\alpha^w}{\pi_\beta C_0 c_\beta^\mu c_\beta^w} = \frac{\pi_\alpha}{\pi_\beta} \exp\left(\Psi_\mu(\alpha, \beta) + \Psi_{\boldsymbol{w}}(\alpha, \beta)\right). \tag{6}$$

In the Appendix H.1 of Lin & Lee (2024), they have proved that when the context length $T \to \infty$, under the first condition in Assumption E.2, $\lim_{T\to\infty} \Psi_\mu(\alpha, \beta) = \geq 0$.

Now we prove that when the empirical risk on the in-context example of pretraining task function $\alpha$ is no more than that of $\beta$, the second term $\Psi_{\boldsymbol{w}}(\alpha, \beta) \geq 0$.

$$\Psi_{\boldsymbol{w}}(\alpha, \beta)$$

$$= \log\left(\frac{\exp\left(-\frac{\|\boldsymbol{w}_\alpha\|^2 - \|\boldsymbol{w}_\alpha + T\delta_w \overline{\boldsymbol{w}}\|^2_{(\boldsymbol{I}+T\delta_w \boldsymbol{\Sigma}_{\boldsymbol{w}})^{-1}}}{2\sigma_w^2}\right)}{\exp\left(-\frac{\|\boldsymbol{w}_\beta\|^2 - \|\boldsymbol{w}_\beta + T\delta_w \boldsymbol{w}\|^2_{(\boldsymbol{I}+T\delta_w \overline{\boldsymbol{\Sigma}}_{\boldsymbol{w}})^{-1}}}{2\sigma_w^2}\right)}\right)$$

$$= \frac{\|\boldsymbol{w}_\beta\|^2 - \|\boldsymbol{w}_\beta + T\delta_w \overline{\boldsymbol{w}}\|^2_{(\boldsymbol{I}+T\delta_w \overline{\boldsymbol{\Sigma}}_{\boldsymbol{w}})^{-1}}}{2\sigma_w^2} - \frac{\|\boldsymbol{w}_\alpha\|^2 - \|\boldsymbol{w}_\alpha + T\delta_w \overline{\boldsymbol{w}}\|^2_{(\boldsymbol{I}+T\delta_w \overline{\boldsymbol{\Sigma}}_{\boldsymbol{w}})^{-1}}}{2\sigma_w^2}$$

$$= \frac{\|\boldsymbol{w}_\beta\|^2 - \left\|\boldsymbol{w}_\beta + \delta_w \sum_{i=1}^{T} \boldsymbol{x}_i y_i\right\|^2_{(\boldsymbol{I}+T\delta_w \overline{\boldsymbol{\Sigma}}_{\boldsymbol{w}})^{-1}}}{2\sigma_w^2} - \frac{\|\boldsymbol{w}_\alpha\|^2 - \left\|\boldsymbol{w}_\alpha + \delta_w \sum_{i=1}^{T} \boldsymbol{x}_i y_i\right\|^2_{(\boldsymbol{I}+T\delta_w \overline{\boldsymbol{\Sigma}}_{\boldsymbol{w}})^{-1}}}{2\sigma_w^2}$$

$$= \frac{\|\boldsymbol{w}_\beta\|^2 - \left\|(\boldsymbol{w}_\beta - \frac{\sum_{i=1}^{T} \boldsymbol{x}_i y_i}{T}) + (\boldsymbol{I} + T\boldsymbol{I}\delta_w)\frac{\sum_{i=1}^{T} \boldsymbol{x}_i y_i}{T}\right\|^2_{(\boldsymbol{I}+T\delta_w \overline{\boldsymbol{\Sigma}}_{\boldsymbol{w}})^{-1}}}{2\sigma_w^2}$$

$$- \frac{\|\boldsymbol{w}_\alpha\|^2 - \left\|(\boldsymbol{w}_\alpha - \frac{\sum_{i=1}^{T} \boldsymbol{x}_i y_i}{T}) + (\boldsymbol{I} + T\boldsymbol{I}\delta_w)\frac{\sum_{i=1}^{T} \boldsymbol{x}_i y_i}{T}\right\|^2_{(\boldsymbol{I}+T\delta_w \overline{\boldsymbol{\Sigma}}_{\boldsymbol{w}})^{-1}}}{2\sigma_w^2}$$

$$\overset{(a)}{=} \|\boldsymbol{w}_\beta - \frac{\sum_{i=1}^{T} \boldsymbol{x}_i y_i}{T}\|^2_{\boldsymbol{I}-(\boldsymbol{I}+T\delta_w \overline{\boldsymbol{\Sigma}}_{\boldsymbol{w}})^{-1}} - \|\boldsymbol{w}_\alpha - \frac{\sum_{i=1}^{T} \boldsymbol{x}_i y_i}{T}\|^2_{\boldsymbol{I}-(\boldsymbol{I}+T\delta_w \overline{\boldsymbol{\Sigma}}_{\boldsymbol{w}})^{-1}}$$

$$\overset{(b)}{=} \|\boldsymbol{w}_\beta - \frac{\sum_{i=1}^{T} \boldsymbol{x}_i y_i}{T}\|^2_{\frac{\delta_w \sum_{i=1}^{T} \boldsymbol{x}_i \boldsymbol{x}_i^\top}{1+\delta_w \sum_{i=1}^{T} \boldsymbol{x}_i^\top \boldsymbol{x}_i}} - \|\boldsymbol{w}_\alpha - \frac{\sum_{i=1}^{T} \boldsymbol{x}_i y_i}{T}\|^2_{\frac{\delta_w \sum_{i=1}^{T} \boldsymbol{x}_i \boldsymbol{x}_i^\top}{1+\delta_w \sum_{i=1}^{T} \boldsymbol{x}_i^\top \boldsymbol{x}_i}}$$

$$\tag{7}$$

where equation $(a)$ is due to the third condition in Assumption E.2, equation $(b)$ is by applying the Sherman–Morrison formula. Since $\frac{\delta_w}{1+\delta_w \sum_{i=1}^{T}} \geq 0$, to prove that $\Psi_{\boldsymbol{w}}(\alpha, \beta) \geq 0$, we only need to show that

$$\|\boldsymbol{w}_\beta - \frac{\sum_{i=1}^{T} \boldsymbol{x}_i y_i}{T}\|^2_{\sum_{i=1}^{T} \boldsymbol{x}_i \boldsymbol{x}_i^\top} - \|\boldsymbol{w}_\alpha - \frac{\sum_{i=1}^{T} \boldsymbol{x}_i y_i}{T}\|^2_{\sum_{i=1}^{T} \boldsymbol{x}_i \boldsymbol{x}_i^\top} \geq 0. \tag{8}$$

Further, we can derive that the term $\frac{1}{T}\sum_{i=1}^{T} \|\boldsymbol{w}_\beta - \boldsymbol{x}_i y_i\|^2_{\boldsymbol{x}_i \boldsymbol{x}_i^T} - \|\boldsymbol{w}_\alpha - \boldsymbol{x}_i y_i\|^2_{\boldsymbol{x}_i \boldsymbol{x}_i^T}$ below is non-negative by using the condition 2 in Assumption E.2:

$$
\frac{1}{T}\sum_{i=1}^{T} \|\boldsymbol{w}_\beta - \boldsymbol{x}_i y_i\|^2_{\boldsymbol{x}_i \boldsymbol{x}_i^T} - \|\boldsymbol{w}_\alpha - \boldsymbol{x}_i y_i\|^2_{\boldsymbol{x}_i \boldsymbol{x}_i^T}
$$

$$
= \frac{1}{T}\sum_{i=1}^{T} (\boldsymbol{w}_\beta - \boldsymbol{x}_i y_i)^\top \boldsymbol{x}_i \boldsymbol{x}_i^T (\boldsymbol{w}_\beta - \boldsymbol{x}_i y_i) - (\boldsymbol{w}_\alpha - \boldsymbol{x}_i y_i)^\top \boldsymbol{x}_i \boldsymbol{x}_i^T (\boldsymbol{w}_\alpha - \boldsymbol{x}_i y_i)
$$

$$
= \frac{1}{T}\sum_{i=1}^{T} (\boldsymbol{w}_\beta + \boldsymbol{w}_\alpha - 2\boldsymbol{x}_i y_i)^\top \boldsymbol{x}_i \boldsymbol{x}_i^T (\boldsymbol{w}_\beta - \boldsymbol{w}_\alpha) \tag{9}
$$

$$
\underset{(c)}{\geq} \frac{1}{T}\sum_{i=1}^{T} (\boldsymbol{w}_\beta + \boldsymbol{w}_\alpha - 2\boldsymbol{x}_i y_i)^\top (\boldsymbol{w}_\beta - \boldsymbol{w}_\alpha)
$$

$$
= \frac{1}{T}\sum_{i=1}^{T} \|\boldsymbol{w}_\beta^\top \boldsymbol{x}_i - y_i\|^2 - \|\boldsymbol{w}_\alpha^\top \boldsymbol{x}_i - y_i\|^2 \underset{(d)}{\geq} 0
$$

where the inequality $(c)$ holds since according to the condition 2 in Assumption E.2, $\boldsymbol{x}_i \boldsymbol{x}_i^T - \boldsymbol{I}$ is positive semi-definite, and the inequality $(d)$ holds since the population downstream risk of $\alpha$ is lower than $\beta$. Therefore, to prove inequality (8), we just need to prove that the l.h.s. of inequality (8) multiplying $\frac{1}{T}$ is not less than $\frac{1}{T}\sum_{i=1}^{T} \|\boldsymbol{w}_\beta - \boldsymbol{x}_i y_i\|^2_{\boldsymbol{x}_i \boldsymbol{x}_i^T}$ in Equation (9):

$$
\frac{1}{T}\left( \|\boldsymbol{w}_\beta - \frac{\sum_{i=1}^{T}\boldsymbol{x}_i y_i}{T}\|^2_{\sum_{i=1}^{T}\boldsymbol{x}_i \boldsymbol{x}_i^\top} - \|\boldsymbol{w}_\alpha - \frac{\sum_{i=1}^{T}\boldsymbol{x}_i y_i}{T}\|^2_{\sum_{i=1}^{T}\boldsymbol{x}_i \boldsymbol{x}_i^\top} \right) \geq \frac{1}{T}\sum_{i=1}^{T} \|\boldsymbol{w}_\beta - \boldsymbol{x}_i y_i\|^2_{\boldsymbol{x}_i \boldsymbol{x}_i^T} - \|\boldsymbol{w}_\alpha - \boldsymbol{x}_i y_i\|^2_{\boldsymbol{x}_i \boldsymbol{x}_i^T}.
$$
$$\tag{10}$$

First, let's simplify the l.h.s of inequality (10):

$$
\frac{1}{T}\left( \|\boldsymbol{w}_\beta - \frac{\sum_{i=1}^{T}\boldsymbol{x}_i y_i}{T}\|^2_{\sum_{i=1}^{T}\boldsymbol{x}_i \boldsymbol{x}_i^\top} - \|\boldsymbol{w}_\alpha - \frac{\sum_{i=1}^{T}\boldsymbol{x}_i y_i}{T}\|^2_{\sum_{i=1}^{T}\boldsymbol{x}_i \boldsymbol{x}_i^\top} \right)
$$

$$
= \frac{1}{T}\sum_{i=1}^{T} (\boldsymbol{w}_\beta - \frac{\sum_{j=1}^{T}\boldsymbol{x}_j y_j}{T})^\top \boldsymbol{x}_i \boldsymbol{x}_i^\top (\boldsymbol{w}_\beta - \frac{\sum_{j=1}^{T}\boldsymbol{x}_j y_j}{T}) - (\boldsymbol{w}_\alpha - \frac{\sum_{j=1}^{T}\boldsymbol{x}_j y_j}{T})^\top \boldsymbol{x}_i \boldsymbol{x}_i^\top (\boldsymbol{w}_\alpha - \frac{\sum_{j=1}^{T}\boldsymbol{x}_j y_j}{T})
$$

$$
= \frac{1}{T}\sum_{i=1}^{T} \|\boldsymbol{w}_\beta^\top \boldsymbol{x}_i - \frac{1}{T}\sum_{j=1}^{T}\boldsymbol{x}_j^\top \boldsymbol{x}_i y_j\|^2 - \|\boldsymbol{w}_\alpha^\top \boldsymbol{x}_i - \frac{1}{T}\sum_{j=1}^{T}\boldsymbol{x}_j^\top \boldsymbol{x}_i y_j\|^2
$$

$$
= \frac{1}{T}\sum_{i=1}^{T} (\boldsymbol{w}_\beta^\top \boldsymbol{x}_i)^2 - (\boldsymbol{w}_\alpha^\top \boldsymbol{x}_i)^2 + 2(\boldsymbol{w}_\alpha - \boldsymbol{w}_\beta)^\top \boldsymbol{x}_i \frac{1}{T}\sum_{j=1}^{T}\boldsymbol{x}_j^\top \boldsymbol{x}_i y_j.
$$
$$\tag{11}$$

Then we simplify the r.h.s. of inequality (10):

$$
\frac{1}{T}\sum_{i=1}^{T} \|\boldsymbol{w}_\beta - \boldsymbol{x}_i y_i\|^2_{\boldsymbol{x}_i \boldsymbol{x}_i^T} - \|\boldsymbol{w}_\alpha - \boldsymbol{x}_i y_i\|^2_{\boldsymbol{x}_i \boldsymbol{x}_i^T}
$$
$$\tag{12}$$
$$
= \frac{1}{T}\sum_{i=1}^{T} (\boldsymbol{w}_\beta^\top \boldsymbol{x}_i)^2 - (\boldsymbol{w}_\alpha^\top \boldsymbol{x}_i)^2 + 2(\boldsymbol{w}_\alpha - \boldsymbol{w}_\beta)^\top \boldsymbol{x}_i \boldsymbol{x}_i^\top \boldsymbol{x}_i y_i
$$

Subtracting Equation (12) from Equation (11), we get

$$
\frac{1}{T}\sum_{i=1}^{T} 2(\boldsymbol{w}_\alpha - \boldsymbol{w}_\beta)^\top \boldsymbol{x}_i \frac{1}{T}\sum_{j=1}^{T}\boldsymbol{x}_j^\top \boldsymbol{x}_i y_j - 2(\boldsymbol{w}_\alpha - \boldsymbol{w}_\beta)^\top \boldsymbol{x}_i \boldsymbol{x}_i^\top \boldsymbol{x}_i y_i
$$
$$\tag{13}
$$
$$
= \frac{1}{T}\sum_{i=1}^{T} 2(\boldsymbol{w}_\alpha - \boldsymbol{w}_\beta)^\top \boldsymbol{x}_i y_i \frac{1}{T}\sum_{j=1}^{T}\left( \boldsymbol{x}_j^\top \boldsymbol{x}_i \frac{y_j}{y_i} - \boldsymbol{x}_i^\top \boldsymbol{x}_i \right).
$$

applying the condition 4 in Assumption E.2, we get the final conclusion.

□

## F    LIMITATIONS

1) Most experimental results are based on a GPT-2 model pretrained on a limited set of mathematical functions. It is challenging to assess whether modern large-scale language models like GPT-4 and Claude 3 Opus face similar difficulties in generalizing beyond their pretraining corpus, given the vast range of tasks and content in their pretraining data. Nevertheless, our findings highlight the limitations of ICL in solving challenging tasks for smaller models like Llama-2-7B and Llama-3-8B. 2) The models are trained on ICL data, while real-world LLMs are trained autoregressively. However, the ICL pretraining objective is also next-token prediction, so there is no clear gap between these two pretraining objectives.

## G    REPRODUCIBILITY

In the main text and Appendix C, we've stated all setups for reproducing our experimental results. For the theoretical part, we've included the assumptions (Assumption E.2) and proofs in Appendix E.

