# OpenReview forum: "Can In-context Learning Really Generalize to Out-of-distribution Tasks?"
_ICLR.cc/2025/Conference — ICLR 2025 Poster_

### Official Review · Reviewer_gxdx · 2024-10-17

**Soundness:** 3
**Presentation:** 2
**Contribution:** 1
**Rating:** 5
**Confidence:** 4

**Summary:**

This paper studied OoD performance with in-context learning in Transformers. The notion of OoD was both with respect to the ground-truth data-generating process at inference time and the distribution of inputs. The main finding is that Transformers do not generalize in these OoD settings, and instead make predictions in accordance with the in-distribution task from the pre-training distribution that achieves the lowest test error on the novel OoD task. This core finding holds when considering simple regression tasks (e.g., linear and quadratic regression), retrieval tasks, and mixtures of tasks.

**Strengths:**

1. The experimental setups are quite easily to follow and well-explained.
2. The tasks introduced in Section 3.3 are quite interesting, as they constitute a simple and meaningful task that we would want a model to generalize to, but we are confident that no such task could have accidentally appeared in the training data.

**Weaknesses:**

1. The core results are to a large extent unsurprising, and don’t clearly tell us anything new. Transformers are essentially just deep neural network (DNN) sequence models, and we know that DNNs don’t generalize out-of-distribution in the absence of the right kinds of inductive biases. Why, then, would we expect a Transformer to generalize in the OoD settings studied in this paper? For instance, would anyone have ever genuinely expected a Transformer trained on (x,y) sequences with a linear relationship to then generalize to (x,y) sequences with quadratic relationships? I expect that the answer is no, and that almost everyone familiar with the deep learning literature will be unsurprised by the empirical findings in Sections 2, 3, and 4. Lack of unexpected results is my primary critique of this paper.
2. This work did not engage with the dominant theory of ICL: that it does Bayes-optimal prediction. This is particularly surprising, since the Bayes-optimal prediction framework seems to account for all of the results. To account for Section 2: the Transformer learns a prior over the pre-training tasks and so when confronted with a new one, the best it can do is implement the function with non-zero prior that best explains the novel OoD data. To account for Section 4: the learned task prior includes a prior over task classes, and given some context the posterior is higher over the tasks that are consistent with that context.
3. Several typos and ungrammatical sentences throughout the paper that make it more difficult to read. For example, there is even a typo in a subsection title “3.3 REAL-WORLD LLMS MAT NOT NECESSARILY IN-CONTEXT LEARN NEW TASKS”. The paper requires a thorough proofread.

**Questions:**

1. Which findings in particular do the authors believe are genuinely surprising to most deep learning researchers?
2. What findings from the paper can’t be explained through the theoretical Bayes-optimal prediction framework for ICL?

---

> ### Author Response · Authors · 2024-11-22
> **Response to Review gxdx (part 1)**
>
> We thank Reviewer gxdx for your careful reading and valuable comments! We'd like to address your concerns on the following points:
>
> ****
>
> Response to Weakness 1: Although it might seem natural that Transformers cannot achieve OOD generalization without appropriate inductive biases, however:
>
> * **ICL is not conventional learning, whether ICL can solve OOD tasks remains controversial. ** Wealth of empirical evidence has demonstrated the emergence of generalization capabilities in real-world LLMs with ICL across various tasks that are unlikely to have been seen during pretraining. These capabilities include performing classification tasks with meaningless labels [1], fitting arbitrary non-linear functions [2], translating very rare languages [3], and classifying writing styles on private datasets not included in the pretraining corpus [4]. **Therefore, it remains unclear whether the ability of ICL to solve these seemingly novel tasks represents true OOD generalization or is simply because LLMs have been exposed to similar tasks during pretraining.** This OOD issue in ICL is considered "an important problem with potential impact" by reviewer 7eJd and "has not been carefully demonstrated yet" by reviewer MhX7. Our work addresses this uncertainty by conducting experiments with rigorously controlled pretraining and test distributions, indicating that ICL may struggle to learn arbitrary new tasks, challenging previous literature [1-4].
>
> Here we summarize our **novel findings and insights**:
>
> * to the best of our knowledge, we are the first to demonstrate that the OOD behavior of ICL closely resembles finding a near-optimal solution within the pretraining distribution through experiments with deep Transformers (Sec. 2).
> * we find that being able to classify OOD abstract labels is a capability of retrieval rather than learning OOD input-output relationships (Sec. 3.1), challenging the findings suggested by [4]. Further, such retrieval ability only emerge when the underlying task is ID (Sec. 3.2).
> * we reveal that the algorithm selection mechanism consistently exist regardless of the downstream distribution and demonstrate how ICL selects its pretraining prior based on the test context (Sec. 4).
>
> ****
>
> Response to Weakness 2: Thank you for highlighting the connection between our work and the existing Bayes-optimal theory for ICL. While several studies attempt to understand ICL from a Bayesian perspective ([5-9]), these works have certain limitations that may reduce their practical applicability:
>
> * **Limited empirical verification.** [5] and [7] lack empirical verification of their theories on real deep transformer models. [9] claims that training on sufficiently many linear regression vectors can cause ICL to generalize to the test distribution, which is considered a non-Bayesian property. However, the distributional shifts in their experiments are not significant (see Appendix A.3 for details) enough to conclude that ICL can truly generalize.
> * **Assumption of in-distribution downstream tasks.** [5] assumes that downstream tasks are components of the pretraining distribution. Similarly, [6] assumes that the latent concept of the test task $\theta^*$ is within the pretraining task set $\Theta$. In [8], both training and testing tasks are linear regression problems with weights sampled from Gaussian distributions.
> * **Limited implications of the theoretical results:** Although [6] and [7] prove that ICL can infer a latent task concept $\theta$ based on the downstream test context $S_{\text{test}}$, they do not reveal how $S_{\text{test}}$ concretely affects the posterior distribution $P(\theta \mid S_{\text{test}})$ inferred by the model, which determines the downstream ICL prediction—especially when the true downstream task $\theta^*$ is OOD.
>
> Our work addresses the aforementioned limitations through the following contributions:
>
> * **Comprehensive empirical evidence.** Our findings are supported by extensive experiments, including synthetic tasks evaluated on deep GPT-2 and natural-language tasks tested on pretrained Llama-2 and Llama-3 models (newly added in Sec. 2.3 and 4.3). These experiments validate and extend the results of [5-7] to broader, more practical scenarios.
> * **More realistic settings reflective of real-world tasks.** We explore OOD tasks where the test function classes differ significantly from the training ones, better mirroring real-world challenges. The results presented in Sec. 2.2 (newly added) suggest a revision of findings from [9], showing that ICL cannot generalize to OOD tasks even after extensive training on ID tasks.
> * **Novel insights into ICL's capabilities.** Our analysis uncovers how the downstream and pretraining distributions influence ICL predictions (Sec. 4). Additionally, we provide a new perspective on the widely observed phenomenon of abstract-label learning, showing that it reflects a retrieval capability rather than an ability to learn unseen tasks.

---

> ### Author Response · Authors · 2024-11-22
> **Response to Review gxdx (part 2)**
>
> Response to Weakness 3: We sincerely apologize for any typos or grammatical errors in the previous version of our paper and regret any inconvenience they may have caused during your reading. We have now thoroughly revised the manuscript.
>
> ****
>
> Response to Question 1: We believe the most insightful finding of our work is as follows: **Despite the impressive empirical performance of real-world LLMs in solving seemingly novel tasks through ICL, we observe that when faced with an OOD task, ICL operates by identifying the most suitable pretraining meta-distribution based on test error and input distribution discrepancies, and th·en attempts to find an optimal solution within that meta-distribution. Notably, this process occurs independently of the downstream test distribution.** Our work provides grounded empirical and theoretical evidence to clarify both the capabilities and limitations of ICL, which we believe offers valuable insights for the community in understanding the boundaries of ICL's potential.
>
> ****
>
> Response to Question 2: We acknowledge that the existing Bayes-optimal prediction framework can explain the findings in Sec.2 that ICL implements in-distribution predictions. However:
>
> * The framework cannot account for **the emergence of OOD retrieval ability** in Sec. 3.1, where ICL retrieves OOD labels after training on a sufficient number of retrieval tasks. Bayes-optimal predictors are not inherently designed to generalize to novel input-label relationships.
>
> * The framework **lacks insight into algorithm selection for OOD contexts.** While classic Bayesian theories, such as [10], have demonstrated similar selection mechanisms for mixed Gaussian models, they fall short in predicting the behavior of ICL performed by complex deep Transformers in OOD tasks. Moreover, existing Bayesian frameworks ([5-9]) have not identified the factors that influence ICL's algorithm selection in arbitrary OOD contexts.
>
>   Our work addresses this gap by characterizing the specific factors that affect downstream ICL predictions (Sec. 4). Furthermore, our theory is notably general: unlike [11-12], we do not assume that test tasks are seen during training, nor does our proof rely on manually constructed model parameters as in [11-12].
>
> We have added detailed discussions to Appendix A.3.
>
> ****
>
> [1] Disentangling task recognition and task learning, Pan et al., Master’s thesis, Princeton University, 2023
>
> [2] From words to numbers: Your large language model is secretly a capable regressor when given in-context examples, Vacareanu et al., Arxiv 2024
>
> [3] Many-Shot In-Context Learning, Agarwal et al., Arxiv 2024.
>
> [4] In-context learning learns label relationships but is not conventional learning, Kossen et al., ICLR 2024
>
> [5] The learnability of in-context learning, NeurIPS 2023
>
> [6] An explanation of in-context learning as implicit Bayesian inference, Xie et al., Arxiv 2021
>
> [7] What and how does in-context learning learn? Bayesian model averaging, parameterization, and generalization., Zhang et al., Arxiv 2023
>
> [8] Dual operating modes of in-context learning, Lin et al., ICML 2024
>
> [9] Pretraining task diversity and the emergence of non-bayesian in-context learning for regression, Raventós, et al., NeurIPS 2023
>
> [10] Pattern recognition and machine learning. Bishop, C. M. (2006). Springer. Chapter 9
>
> [11] Transformers as statisticians: Provable in-context learning with in-context algorithm selection, Bai et al., NeurIPS 2023
>
> [12] In-context learning on function classes unveiled for transformers, Wang et al., ICML 2024

---

> > ### Comment · Reviewer_gxdx · 2024-11-22
> >
> > Thank you for the responses and for meaningfully engaging with my feedback.
> >
> > **Bayesian interpretation**
> >
> > I would like to continue challenging the point that the Bayesian framework does not account for you results. For instance, in your response you claimed that:
> >
> > >we observe that when faced with an OOD task, ICL operates by identifying the most suitable pretraining meta-distribution based on test error and input distribution discrepancies, and then attempts to find an optimal solution within that meta-distribution. Notably, this process occurs independently of the downstream test distribution
> >
> > Here's my understanding of the Bayesian reframing, which I believe reflects exactly your interpretation above. During pretraining, the model has learned a prior over task variables, an inference mechanism to infer the posterior over those task variables from prior data (for simplicity, let's say that it does MAP), and a likelihood function to predict the next-token given the inferred task variable. At test time when you present and OOD task, the model performs (for a supervised in-context task):
> >
> > $p(y_t|x_t) = \arg\max_\theta p(y_t | x_t, \theta) p(\theta | x_{1:t-1})$
> >
> > Now here's how I explain the results (which again, I see as identical to your quote above). The posterior $p(\theta | x_{1:t-1})$ of course can't settle on the true latent variable for the OOD task because it assigns $\approx 0$ prior to it (it never saw such a variable at training time). The best it can do is use the solution that has a reasonable posterior $p(\theta | x_{1:t-1})$ (which will only put probability mass on task variables within the pretraining meta-distribution) while doing the best it could to explain the test data $p(y_t | x_t, \theta)$.
> >
> > Here are my questions to you:
> > 1. Have I missed something in the argument above, or would you agree that the Bayesian interpretation is both consistent with your results, and actually is just a more formal rewording of the explanations you're already giving for those results?
> > 2. If you believe the Bayesian interpretation is the same one as what you're providing, why not reframe the entire paper as an empirical evaluation of predictions that the Bayesian theory makes for what would happen on OOD data?
> >
> > By the way, regarding (2),  you have convinced me by the way that your paper provides such an empirical evaluation that is missing in the Bayesian ICL literature. I therefore agree that it has empirical value, but I would not frame these as novel theoretical findings. Instead, I would frame it as an empirical validation of predictions made by the Bayesian theory for the OOD task case. In other words, the results are "novel" in how they support the Bayesian theory; while they may not be surprising to people who hold the Bayesian view, they provide independent support of it.
> >
> > **Association/retrieval and algorithm selection**
> >
> > Apologies for sounding like a broken record, but I still see these results as natural consequences of the Bayesian view of ICL. For association, the "task variable" is just a similarity function between $x$ tokens that can be learned from context. The task variable doesn't include the $y$ id (and so it isn't OOD); it only include the similarity function between $x$'s and a history of all past $(x,y)$ pairs, then retrieves the subsequent $y$ tokens following past $x$ tokens that $\theta$ says are similar to the current one. When the true test-time similarity function $\theta_{test}$ is OOD, then ICL fails (as the Bayesian view predicts it should).
> >
> > For the algorithm selection, the task variable just includes a distribution over function classes. Again, the Bayesian view predicts that the posterior over this function class distribution (and the parameters of the function in that class) will govern test-time behaviour.
> >
> > Have I missed something here? Don't we have to just specify the correct semantics for the task variable $\theta$ in order to explain these cases using the Bayesian perspective? If so, I fail to see how they are fundamentally different kinds of phenomena that require different explanations for OOD behaviour, and therefore don't constitute independent contributions.

---

> > > ### Author Response · Authors · 2024-11-23
> > > **Further response to Reviewer gxdx**
> > >
> > > Thank you for your insightful interpretation of the algorithm selection and OOD retrieval ability from a Bayesian perspective.
> > >
> > > However, we note that this work's main focus is to clarify **our understanding of real-world Transformers on OOD tasks** through rigorous ablation studies, instead of verifying one type of ICL interpretation. As illustrated in Appendix A.1 and A.2, other ICL theories (there are many in the literature, such as the gradient descent perspective [13-14], construction of Transformer weights [15]) could also potentially explain these phenomena. Meanwhile, we fully resonate with you that the Bayesian interpretation provides valuable insights, and we have added a section in Appendix A.4 to elaborate on this connection. We hope that our empirical observations could inspire more theoretical interests looking into this problem.
> > >
> > > [13] Trained Transformers Learn Linear Models In-Context, Zhang et al., JMLR 2024
> > >
> > > [14] On Mesa-Optimization in Autoregressively Trained Transformers: Emergence and Capability, Zheng et al., NeurIPS 2024
> > >
> > > [15] In-context Learning on Function Classes Unveiled for Transformers, Wang et al., ICML 2024

---

> ### Author Response · Authors · 2024-11-27
> **Gentle Reminder for Your Valuable Feedback**
>
> Dear reviewer gxdx,
>
> I hope this message finds you well. I wanted to kindly follow up regarding our previous response to your comments. If there are any further questions or clarifications needed, we would be more than happy to provide additional information or explanations to assist in the review process. Thank you for your time and consideration.

---

> > ### Comment · Reviewer_gxdx · 2024-11-27
> >
> > Apologies for my late reply. Given that the reviewers have engaged with my feedback and added some connections to the Bayesian interpretation of ICL, I have adjusted my score to a 5. The reason that I don't want to adjust it higher than that is because of what I've said earlier: (1) the paper is essentially an empirical confirmation of the Bayesian interpretation of ICL, (2) I don't believe the results will surprise people who were already on board with this interpretation, and (3) I don't find the paper adequately frames the findings in this Bayesian perspective (which I think would have been cleaner than the current language used to explain the findings, which are less unified under a single theoretical interpretation as the paper is currently written).
> >
> > In sum, as an empirical paper studying ICL, I think it would be much stronger if structured as:
> > 1. Here's some theoretical frameworks for understanding what ICL is doing
> > 2. Here are the different predictions they make about OoD generalization on tasks a, b, etc.
> > 3. The results support theory X
> >
> > In contrast, the current paper is written is more structured as:
> > 1. Here's some ICL tasks and OoD settings
> > 2. Here are the results
> > 3. On task (a) the model generalizes in this way, on task (b) the model generalizes in this way, ...
> > 4. Overall, the model uses the closest function in its training distribution that minimizes OoD error (without strongly relating this to theories that already predict such behaviour)
> >
> > The results could still be interesting to some even if they did not surprise me of course, and in that regard I think the paper has value, but I strongly believe that the first style I suggested above for framing the work and results would have been far more effective. In this context, I think my current score of 5 accurately reflects my view of the work, and if the paper does not get accepted here, I encourage the authors to try and contextualize their findings in the language of dominant theoretical paradigms for ICL in future submissions.

---

> > > ### Author Response · Authors · 2024-12-01
> > > **Further response to Reviewer gxdx**
> > >
> > > We would like to thank reviewer gxdx for raising the score and for suggesting ways to improve our work!
> > >
> > > Following your advice, we have revised our paper by adding a new section preceding the original Sec. 2 to discuss existing theoretical predictions regarding the behavior of ICL. However, as we are currently unable to modify the PDF, we briefly quote some content from this newly added section below,
> > >
> > > "Previous literature have provided some theoretical insights into the behavior of ICL. Here we briefly review some of the representative findings. **1) ICL makes Bayesian predictions.** ... **2) ICL implements gradient descents.** ... **3) ICL implements algorithm selection.** ... **4) ICL performs retrieval.** ...  These theoretical findings may appear disparate, as they describe different aspects of ICL under varying assumptions and settings. Furthermore, most of them remain largely unexplored empirically, particularly on large-scale nonlinear Transformers. ..."
> > >
> > > We also connect our empirical findings in each section to existing theories, discussing how our experimental results validate or expand upon prior theoretical frameworks.
> > >
> > > Finally,  we would like to express our gratitude to the reviewer gxdx for your thoughtful and meticulous feedback, as well as for the time and effort you have dedicated to improving our work.

---

> > > > ### Comment · Reviewer_gxdx · 2024-12-02
> > > >
> > > > Thank you for continuing to consider my feedback. At this point, to raise my score I think I would unfortunately need to read the paper as a whole with the changes you’ve made. I think to raise it without seeing how it reads now and the way the empirical results are put into context of ICL theories would be irresponsible. I will, however, flag to the AC here that these sorts of changes made by the authors are precisely the sort of thing that I think would make this a strong and interesting empirical paper; I will leave the rest to the AC’s judgement.

---

### Official Review · Reviewer_7eJd · 2024-11-01

**Soundness:** 2
**Presentation:** 2
**Contribution:** 2
**Rating:** 6
**Confidence:** 3

**Summary:**

The paper conducts a series of empirical studies on synthetic tasks aimed to characterize the ability of transformers to generalize to unseen function families using in-context learning (ICL). It finds that while ICL allows generalization to new tasks from known function families, it fails to generalize to tasks from unseen function families instead predicting solutions in terms of the known function families encountered during training that achieve the lowest test-error. It further demonstrates that the strong in-context retrieval ability of transformers can sometimes give a misleading impression of out-of-distribution generalization.

**Strengths:**

- Characterizing the out-of-distribution generalization ability of in-context learning is an important problem with potential impact
- The experiments are well-designed to investigate the respective hypotheses
- The paper does a good job at referencing and contextualizing existing related work

**Weaknesses:**

- The presentation of the paper could be improved, in particular I found section 4.2 hard to follow (see questions) and with the many different experiments often following prior work it was difficult to identify what findings were novel and specific to this paper.
- The theoretical results heavily borrow from [1] (but the paper makes this very transparent)

[1] Lin, Ziqian, and Kangwook Lee. "Dual operating modes of in-context learning." arXiv preprint arXiv:2402.18819 (2024).

**Questions:**

- It is not clear to me whether the Llama2 OOD synthetic word classification task in section 3.3 fails due to a lack in OOD generalization or because of other factors that make this task difficult. One control would be to train a model explicitly on this ICL task to see if it is solvable when it is in-distribution. Would it make sense to run such a control?
- I found Section 4.2 and Figure 8 and Figure 9 hard to read and understand. What exactly is shown in the last row of both of these figures? Why are the losses so high here throughout? I am probably misunderstanding this plot but I would have expected to see that the pretrained ICL model achieves the same loss as the corresponding PT task with the lowest test error.
- Would it be possible to show the loss plots on a log scale (especially Figure 7) to get a better understanding how close the different methods are to each other? Many of the plots contain relatively large values (e.g. due to double descent) which make it difficult to assess how close the best models perform at small loss values. Maybe they could be added to the appendix.
- Are the y-ticklabels of Figure 7b+c correct, i.e. is the loss 10x higher for these tasks? If so, why is that the case?

---

> ### Author Response · Authors · 2024-11-22
> **Response to Review 7eJd**
>
> We thank Reviewer 7eJd for your careful reading and valuable comments! We'd like to address your concerns on the following points:
>
> ****
>
> Q1: The presentation of the paper could be improved, in particular I found section 4.2 hard to follow (see questions) and with the many different experiments often following prior work it was difficult to identify what findings were novel and specific to this paper.
>
> A1: Thank you for your feedback! We apologize for any lack of clarity. We have refined the presentation of Section 4.2 and highlighted the novel contributions of each experiment in the updated PDF. Here is a brief summary of the novel findings from all the experiments (all figures below use the indices in the updated PDF):
>
> * Fig. 1: When evaluated on **OOD** tasks, ICL performs similarly to the corresponding model of the pretraining function class optimized by GD given enough in-context examples. This suggests that ICL may lack the capability to learn new functions.
> * Fig. 2 (newly added): The OOD performance cannot improve with training on more diverse (up to $\sim 2^{20}$) ID vectors.
> * Fig. 3: A pretrained Llama-3-8b tends to make in-distribution predictions in a natural-language task.
> * Fig. 4: The retrieval ability can emerge from training on retrieval tasks with more diverse labels.
> * Fig. 5: ICL can perform "predict-then-retrieve" tasks with OOD labels after training on similar tasks with diverse labels. This offers insights into how real-world LLMs develop the capability to perform abstract label classification.
> * Fig. 6: In the "predict-then-retrieve" task, ICL succeeds only if the prediction rule is ID. Otherwise, the model fails even if the target label appears in the context.
> * Fig. 7: A pretrained Llama-3-8b cannot solve the OOD synthetic word classification task, indicating the limits of real-world LLMs in solving OOD tasks through ICL.
> * Fig. 8: Algorithm selection in ICL **broadly happens on OOD tasks**.
> * Fig. 9: The selection of pretraining priors on OOD tasks depends on the test error of the pretraining functions and the distance between training and test inputs.
>
> ****
>
> Q2: The theoretical results heavily borrow from [1] (but the paper makes this very transparent)
>
> A2: We acknowledge that we adopted the theoretical framework from [1] in our main Theorem 4.4 and have provided all necessary credits to [1] in our paper. However, some points need to be clarified:
>
> * **Different Scopes of Analysis**: Their analysis and ours cover different aspects of ICL. [1] focuses on **"task learning"** and **"task recognition"** when both the training and testing functions are linear regression (an in-distribution setting), whereas our analysis concentrates on **how ICL selects pretraining priors when faced with arbitrary OOD functions**.
> * **Non-Trivial Extension of Theory**: While we adopted the closed-form ICL prediction from [1], extending it to derive how the interaction between the training and test distributions affects prior selection is non-trivial.
> * **Novel Insights**: **We extract novel insights from the existing analysis of [1]** (Lemma 4.3 in our paper): ICL prefers pretraining priors with input distributions closer to the test context. However, [1] does not explicitly explore this aspect.
>
> ****
>
> Q3: It is not clear to me whether the Llama2 OOD synthetic word classification task in section 3.3 fails due to a lack in OOD generalization or because of other factors that make this task difficult. One control would be to train a model explicitly on this ICL task to see if it is solvable when it is in-distribution.
>
> A3: Following your advice, to show that the failure in the synthetic word classification task is mainly due to its OOD nature instead of some other factors that make it difficult to learn, we add an experiment in Appendix B.2 in the updated PDF, in which we train a GPT-2 to perform the same task. **We find that the task can be well addressed after training.** In this task, the $\boldsymbol{x}\_i$ and $\boldsymbol{y}\_i$ are generated in the same way as Sec. 3.3. The only modification is that we use a smaller predefined vector embedding $E'\in \mathbb{R}^{10000\times 20}$ instead of the $E\_{llama}\in \mathbb{R}^{32000\times 4096}$ used in the experiment in Sec. 3.3.

---

> ### Author Response · Authors · 2024-11-22
> **Response to Review 7eJd (part 2)**
>
> Q4: What exactly is shown in the last row of both of these figures?
>
> A4: We apologize for the lack of clarity. The final rows of original Fig. 8 and 9 (Fig. 9a and 9b in the updated PDF) present the **test error of the pretrained ICL model using the closed-form prediction derived in Lemma 4.2.**, i.e. $\|\left\langle \boldsymbol{x}\_{i+1}, \sum\_{m=1}^M \tilde{\pi}\_m \tilde{\boldsymbol{w}}\_m\right\rangle-y\_{T+1}^*\|^2$.
>
> ****
>
> Q5: Why are the losses so high here throughout?
>
> A5: The high test loss values can be attributed to the following reasons:
>
> * The test error is calculated using the squared loss without taking the square root.
> * In Figure 8, the absolute values of the centers of the pretraining linear regression weights are large (e.g., $[\pm5,\pm5,\pm5]$), while the test ReLU NN weights are sampled from $\mathcal{N}(0,1)$, which results in relatively smaller magnitudes. This significant numerical discrepancy between the training and test function parameter distributions likely amplifies the gap in their outputs. A similar explanation applies to Figure 9, where the input distributions of the training and test sets differ significantly.
>
> ****
>
> Q6: I am probably misunderstanding this plot but I would have expected to see that the pretrained ICL model achieves the same loss as the corresponding PT task with the lowest test error.
>
> A6: Explanation: According to Lemma 4.2, the closed-form ICL prediction $M^*\left(\mathcal{S}\_i \oplus \boldsymbol{x}\_{i+1}\right)=\left\langle \boldsymbol{x}\_{i+1}, \sum\_{m=1}^M \tilde{\pi}\_m \tilde{\boldsymbol{w}}\_m\right\rangle$ is not the linear combination of the pretraining priors $\boldsymbol{w}\_m$. Instead, it is a combination of $\tilde{\boldsymbol{w}}\_m$, which represents the posterior of $\boldsymbol{w}\_m$ conditioned on the downstream context. As highlighted in [1], the shift from $\boldsymbol{w}\_m$ to $\tilde{\boldsymbol{w}}\_m$ reflects the "task learning" process of ICL, demonstrating that the model can refine its prediction when given the downstream context. This refinement explains why, in Fig. 9, the loss in the last row is lower than that in the first row. The loss in the first row is $\|\left\langle\boldsymbol{x}\_{T+1},\boldsymbol{w}\_m\right\rangle-y\_{T+1}^*\|^2$, while the last row measures the loss of using the refined $\tilde{\boldsymbol{w}}\_m$ to predict, i.e., $\|\left\langle \boldsymbol{x}\_{T+1}, \sum\_{m=1}^M \tilde{\pi}\_m \tilde{\boldsymbol{w}}\_m\right\rangle-y\_{T+1}^*\|^2$.
>
> ****
>
> Q7: Would it be possible to show the loss plots on a log scale (especially Figure 7) to get a better understanding how close the different methods are to each other?
>
> A7: Following your advice, we've rescaled the loss to log scale in Fig. 8 in the updated PDF (original Fig. 7). We can observe that the ICL performance is closest to the best pretraining function class.
>
> ****
>
> Q8: Are the y-tick labels of Figure 7b+c correct
>
> A8: Yes. The loss is large since the value of the labels of OOD functions cubic regression and linear+quadratic regression is high.

---

> ### Author Response · Authors · 2024-11-25
> **Gentle Reminder for Your Valuable Feedback**
>
> Dear Reviewer 7eJd,
>
> We understand that you may be busy and might not have had the opportunity to review our rebuttal yet. With only three days remaining, we kindly remind you once again in the hope of discussing this further with you. We have invested significant time and effort into preparing a detailed response, as your feedback is incredibly important to us. Could you please take a moment to review it? Thank you so much for your time and consideration!

---

> ### Comment · Reviewer_7eJd · 2024-11-25
>
> Thank you for your detailed response. I appreciate the additional experiment you ran and report in Appendix B.2 helping to resolve my concern on the task being generally too difficult. While I believe the presentation of the paper could be further improved by better guiding the reader through the many distinct sections and findings, I remain positive of this work and will maintain my score.
>
> Minor point:
>
> - Thank you for rescaling the loss to a log-scale in Figure 8. May I ask if there is a reason to not consistently do this throughout the paper?

---

> > ### Author Response · Authors · 2024-11-26
> > **Further response to Reviewer 7eJd**
> >
> > Thank you for your recognition of our paper. We sincerely appreciate your valuable suggestions and the time and effort you dedicated to the review process. We will work to further improve our presentation for better clarity.
> >
> > Regarding Fig. 8, we rescaled the loss to a log scale because the loss values in this figure are very close to zero, and using a log scale helps to expand their range for better visualization. For the other figures, however, we found that rescaling to a log scale did not enhance clarity, so we retained the original scale.

---

### Official Review · Reviewer_MhX7 · 2024-11-03

**Soundness:** 4
**Presentation:** 3
**Contribution:** 4
**Rating:** 8
**Confidence:** 4

**Summary:**

This paper investigates the ability of transformer models to generalize to tasks that are OOD compared to its pre-training distribution through in-context learning. They do this through empirical analysis with controlled synthetic tasks and on a pretrained Llama-2-7B model as well as provide some theoretical justification. The core contributions are:
1. The performance of transformers on OOD tasks is similar to the performance of an estimator that learns the ICL examples through GD within the pretrianing function class.
2. The transformers ability to learn and classify abstract concepts in-context can be attributed to its ability to do retrieval and can be learned by doing retrieval on a pre-training distribution of many diverse labels. This fails if the testing function is OOD.
3. Provides theoretical and empirical evidence that when the pretraining distribution contains a mixture of function classes, when tested OOD, models select the pretraining function class that has the lowest test error and match between the train and test input distributions.

**Strengths:**

**Originality**: This paper adds to the literature on understanding ICL in transformers on synthetic tasks by empirically and theoretically characterizing their behavior on OOD tasks. To my knowledge, this behavior on OOD tasks has not been carefully demonstrated yet.

**Quality**: The paper does thorough, well thought out experiments and provides new theoretical insight.

**Clarity**: The paper is clearly written, well structured and easy to understand.

**Significance**:  The question asked by the paper are significant as it further clarifies the mystery of ICL especially regarding limitations to OOD tasks.

**Weaknesses:**

- In section 2, it would be helpful to further clarify what is existing knowledge and what is a new contribution. In fact, I think the presentation would be cleaner if Section 2 and Section 4 were combined into a unified presentation.
- The explanation of the retrieval version of the Llama-2-7B task was hard to understand, it would be helpful to clarify that section
- The conclusion that LLMs cannot learn OOD tasks because its ability to learn abstract concepts from context can be attributed to retrieval felt like a jump in conclusion: see questions).
- I did not understand what hypothesis the experiment in Figure 5 was trying to test and a more clear explanation of why this experiment is useful and what the hypothesis is as an analogy to real world use case would be helpful. It seems like a really whacky OOD task that I am not sure is representative.

**Questions:**

1. In section 4, the authors demonstrate model selection within the pre-training task distribution. Can you connect this to your results in section 2, i.e. does the transformer get the same error as Gradient descent with the function class that gets the lowest error in Figure 7?
2. What causes the sudden increase in performance after context length size 10 in Figure 6B
3. What do you precisely mean by the claim that "the ability of ICL to perform abstract label classification may not serve as an evidence of learning new tasks."? In this abstract label classification task, learning new tasks is the same as being able to classify OOD labels. This is something you show that models are capable of doing if pre-trained on enough diversity of abstract labels tasks (green line in Figure 4c). It seems like this presents evidence that (in a limited context) transformers can learn new OOD tasks.
4. Can you please explain the motivation behind the experiment in Figure 5 better.

---

> ### Author Response · Authors · 2024-11-22
> **Response to Review MhX7**
>
> Thank you for your careful reading and appreciation of our work! We'd like to address your concerns on the following points:
>
> ****
>
> Response to Weakness 1: Our novel findings in Sec. 2: when evaluated on **OOD** tasks, ICL performs comparably to a corresponding model of the pretraining function class optimized by gradient descent (GD), given enough in-context examples. Additionally, we observe a double descent curve when the model is trained on (quadratic) linear regression and evaluated on OOD function classes. These findings suggest that ICL may only optimize within its pretraining function class rather than genuinely learn an OOD function.
>
> Existing findings in Sec. 2: for **ID** tasks, given sufficient in-context examples, the test error can be minimized to near zero.
>
> We have further clarified the novel contributions in Sec. 2 in the updated PDF. While merging Sec. 2 and Sec. 4 is a good suggestion, it would require significant restructuring of the paper, which may not align with the reviewing policy. Therefore, we maintain the current organization of the sections.
>
> ****
>
> Response to Weakness 2: Sorry for the lack of clarity. Here we clarify the task setup of the retrieval task of Llama-2-7b  (Fig. 6(b) in the original PDF, Fig. 13(b) in the new PDF). Concretely, we first randomly sample $C=10$ different vectors from $E_{llama}$ as $\boldsymbol{x}_i$ and compute $\boldsymbol{y}_i$ in the same way as the OOD classification task to get $S=[\boldsymbol{x}_1, \boldsymbol{y}_1,...,\boldsymbol{x}_C, \boldsymbol{y}_C]$. Then we repeat $S$ 10 times to construct the input sequence $S'=[S\oplus S\oplus...\oplus S]$, where $\oplus$ denotes concatenation operation. The goal is to predict the next token given a prefix of $S'$. To succeed in this task, the model has to retrieve the same token as the query token (the last $\boldsymbol{x}_i$ of $S'$) from the context and output $\boldsymbol{y}_i$. We add the clarification and the result of the same experiment using Llama-3-8b in Sec. 3.3. We move the original result of Llama-2-7b to Appendix B.3.
>
> ****
>
> Response to Weakness 3 & Question 3: We do not consider the ability to retrieve OOD labels from the context as a genuine capability to learn OOD tasks because **once the underlying task function is OOD, the model fails even if the target label appears in the context.** This is supported by the experiment in Fig. 6 of the updated PDF (original Fig. 5). In this figure, the test function is OOD, so although the target label integer is likely to present in the context, the model is unable to retrieve it correctly.
>
> Regarding your point that "models are capable of doing so if pre-trained on a diverse range of abstract label tasks," this is valid only when the underlying task is ID, as shown in Fig. 5 of the updated PDF (original Fig. 4). We have clarified it explicitly in Sec. 3.2.
>
> ****
>
> Response to Weakness 4 and Question 4: The experiment in Fig. 6 (original Fig. 5) was designed to evaluate whether ICL can successfully address a task through retrieval when the ground-truth target label is highly likely to appear in the context but the underlying task function is OOD.
>
> **The usefulness of this experiment:** It demonstrates that ICL can only solve a task by retrieving a similar label from the context if the underlying task is ID. When the task function is OOD, retrieval fails to provide a solution.
>
> **Reflections on real-world tasks:** This finding highlights a limitation in improving an LLM's performance through in-context examples. While providing examples with shared labels may seem helpful, this approach may fail if the task is OOD and the underlying prediction rule is too complex for the LLM to learn.
>
> ****
>
> Response to Question 1: Yes. In this experiment, we found that "2-layer ReLU NN" is consistently the most performant training function class across all four OOD test function classes, so we add the error of a 2-layer ReLU NN trained by GD. We find that **the performance of the ReLU model trained by GD aligns well with the ICL performance of the GPT-2 trained on the same function class.** This demonstrates that our findings in Sec. 2 still hold when the transformer is trained on a mixture of tasks.
>
> ****
>
> Response to Question 2: According to the setup explained in Response to Weakness 2, **the length of the repeated subsequence $S$ is 10**, so only after 10 examples can the LLM correctly retrieve the target vector from the context.

---

> > ### Comment · Reviewer_MhX7 · 2024-11-24
> >
> > Thank you for addressing my questions. I also found your responses to the other reviewers informative. I think this is a good paper that adds a novel and useful perspective to ICL. I will maintain my score of 8 and advocate for acceptance.

---

> > > ### Author Response · Authors · 2024-11-24
> > >
> > > Thank you for your recognition! We also sincerely appreciate your valuable suggestions and the time and effort you dedicated during the review process.

---

### Official Review · Reviewer_KJfV · 2024-11-03

**Soundness:** 2
**Presentation:** 2
**Contribution:** 2
**Rating:** 6
**Confidence:** 3

**Summary:**

This paper studied the limitations of in-context learning (ICL) on out-of-distribution (OOD) tasks through empirical experiments and theoretical analysis. They pointed out that ICL mainly implements functions from its pretraining distribution rather than learning new functions. Therefore, ICL cannot guarantee ideal performance when ICL data are from OOD distributions unseen during training. While ICL appears capable of handling tasks with abstract labels, the authors show this ability stems from learning retrieval mechanisms during training rather than true OOD generalization. They also provide theoretical insights into how ICL selects which pretraining function to implement when faced with OOD tasks.

**Strengths:**

1. Novel theoretical insights: The paper provides theoretical analysis of the "low-test-error preference" mechanism in ICL, explaining how models select which pretraining function to implement when handling OOD tasks.
2. I found the experiments on the abstract label classification interesting. The paper provides a fresh perspective on how models handle abstract label classification, demonstrating it's more about retrieval capabilities than true OOD learning.
3. Besides GPT-2, they use Llama2-7b, a pre-trained language model for evaluation.

**Weaknesses:**

1. Presentation issues.

(a) In Eq.1, why the prediction is supervised by f(x_i)? Shouldn’t it be f(x_{I+1})

(b) In Fig.2, what is y_i id? Is it same as I_{y_i}? If yes, why not keep using the same notation? If no, please clarify.

(c)Please correct the way how you are using the quote. See examples in the caption of Fig.3.

2. Questions on the training set. The retrieval design in Sec.3.1 is interesting. When the training range is larger (say yi Id in [50, 455]), the test-time performance is better regardless that y_i in [50, 155] aligns better with the evaluation tasks. I wonder if you have same number of training tasks for different range.  If the answer is no, then is it  a fair comparison?  Is it possible that better performance simply comes more training data? Please clarify the number of training tasks for each range. Second, the setup is hard to understand, please improve the writing.


3. Questions about the experiment design. The first and last experiments in Sec.3 are following the same idea, which maps the index of an embedding to new indices based on some pre-specified rules.  I find it hard to understand the motivation here. Are previous works using similar design? If yes, please cite. If no, could you clarify why this setup?


4. Questions about the assumptions when evaluating pretrained models.In Sec.3.3, how is it possible to know an evaluation task is in-distribution or out-of-distribution for a pretrained llama-2-7b? In line 264, the authors said `To ensure the task is far from the pretraining distribution’, are there evidence supporting the assumption?


5. Limited evaluation setups. All conclusions are derived based on either curve fitting or rule-based mapping. The setups are too simple. They are not sufficient to derive convincing conclusions especially when evaluating the behaviors of models with high parameter size.

**Questions:**

See the comments in the weakness section.


Update:

Sorry for the late response. Most of my concerns were properly addressed. I have raised my score to 6.

---

> ### Author Response · Authors · 2024-11-22
> **Response to Review KJfV**
>
> We thank Reviewer KJfV for your careful reading and detailed comments! We'd like to address your concerns in the following points:
>
> ****
>
> Q1: Presentation issues.
>
> A1: Thank you for meticulously pointing out the presentation issues! We sincerely apologize for these oversights and would like to clarify the revisions we’ve made:
>
> * **In Eq. (1)**, yes, $f(\boldsymbol{x}\_i)$ should be $ f(\boldsymbol{x}\_{i+1}) $.
> * **In Fig. 2**, "$y_i$ id" refers to the index of $y_i$, which corresponds to $I_{y_i}$. We've updated the title of Fig. 2 (Note: Fig. 2 is now Fig. 4 in the updated PDF.)
> * **Quotation marks:** We’ve standardized all quotes by replacing single quotation marks (`'`) with double quotation marks (`"`), ensuring consistency throughout the text.
>
> ****
>
> Q2: I wonder if you have same number of training tasks for different range. ... Second, the setup is hard to understand, please improve the writing.
>
> A2: Yes, all three models, regardless of their respective training ranges for $I_{y_i}$, were trained on 200,000$\times$64 sequences, where 200,000 represents the number of training steps and 64 corresponds to the batch size. The performance gain stems from the model's exposure to more retrieval rules (or equivalently, a larger variety of label vectors).
>
> We sincerely apologize for any inconvenience caused during your reading. We have rewritten Sec. 3.1 to improve clarity and readability, with the revisions highlighted in blue.
>
> ****
>
> Q3: Are previous works using similar design? If yes, please cite. If no, could you clarify why this setup?
>
> A3: To the best of our knowledge, we haven't seen a similar design. The motivation behind repeating a similar synthetic word classification task in Fig. 6(a) (now Fig. 7) is to explore whether the observed phenomenon—that ICL struggles to learn tasks from an unseen distribution—**applies to real-world pretrained LLMs.**
>
> ****
>
> Q4: how is it possible to know an evaluation task is in-distribution or out-of-distribution for a pretrained llama-2-7b? In line 264, the authors said `To ensure the task is far from the pretraining distribution’, are there evidence supporting the assumption?
>
> A4: We acknowledge that it is difficult to determine whether a task was encountered during the pretraining stage of Llama-2. However, Llama-2's pretraining primarily focuses on natural language tasks rather than specialized mathematical or vector-based tasks. The Llama-2 paper [1] also does not mention training on tasks involving learning linear classifiers or vector-to-vector transformations. Therefore, it is likely that the task in Section 3.3—which requires learning a manually generated linear classification function—was unseen during the pretraining of Llama-2.
>
> ****
>
> Q5: Limited evaluation setups.
>
> A5: Following your advice, we add two natural-language experiments with Llama-3-8b to support our findings. Please refer to Sec. 2.3 and Sec. 4.3 of the updated PDF for details.
>
> 1) In Section 2.3, we demonstrate how the tendency of ICL to perform in-distribution (ID) predictions manifests in real-world LLMs. To this end, we designed a task involving predicting labels with letters reversed. The basic tasks include outputting antonyms, translating from English to French, etc. All the letters of the original labels are reversed. **We found that in this task, a pretrained Llama-3-8b tends to output the reversed result of the query word rather than first predicting the correct label and then reversing it.** Although both reversal tasks are uncommon, directly outputting the reversed version of a word is relatively more common than first reasoning and then outputting the reversed prediction. Therefore, this result reflects to some extent that LLMs, when performing ICL, are more inclined to make in-distribution predictions.
> 2) In Section 4.3, the LLM is required to perform an ambiguous sentence classification task through ICL. Each sentence can be classified based on one of three aspects: sentiment, type, or location. For example, the sentence "The groundbreaking discovery made by Japanese scientists has revolutionized renewable energy" can be categorized as ("positive", "science", "Asia"). In each ICL sequence, the label words are chosen from one of these aspects and are replaced with abstract strings. For instance, "positive," "neutral," and "negative" are mapped to "RqF," "IwZ," and "SdK," respectively. **We observe that the LLM can identify the most appropriate classification criterion**, verifying the algorithm-selection mechanism of ICL.
>
> ****
>
> [1] Llama 2: Open Foundation and Fine-Tuned Chat Models

---

> ### Author Response · Authors · 2024-11-25
> **Gentle Reminder for Your Valuable Feedback**
>
> Dear Reviewer KJfV,
>
> We understand that you may be busy and might not have had the opportunity to review our rebuttal yet. With only three days remaining, we kindly remind you once again in the hope of discussing this further with you. We have invested significant time and effort into preparing a detailed response, as your feedback is incredibly important to us. Could you please take a moment to review it? Thank you so much for your time and consideration!

---

> ### Author Response · Authors · 2024-11-27
> **Gentle Reminder for Your Valuable Feedback**
>
> Dear reviewer KJfV,
>
> I hope this email finds you well. I wanted to kindly follow up regarding our previous response to your comments. If there are any further questions or clarifications needed, we would be more than happy to provide additional information or explanations to assist in the review process. Thank you for your time and consideration.

---

> ### Author Response · Authors · 2024-12-02
> **Looking Forward to Your Feedback**
>
> Dear Reviewer KJfV,
>
> This is a gentle reminder that there are only three days left before the discussion period concludes. Your input is incredibly valuable to us.
>
> In our previous responses, we have thoroughly addressed all the concerns you raised and have also included two carefully designed natural language experiments to further support our understanding of ICL's mechanism in OOD scenarios.
>
> We sincerely hope to hear your thoughts and feedback on these updates. Thank you again for your time and effort in reviewing our work!

---

### Author Response · Authors · 2024-11-22
**Updates in the manuscript**

Following the suggestion from the reviewers, we have updated a revised manuscript with the main changes highlighted in blue, which are:

1. Add new experiments.
   * In Sec. 2.2, we add an experiment to show that training on diverse (around $2^{20}$) in-distribution (linear regression) functions cannot improve the OOD performance of ICL.
   * In Sec. 2.3, we add a OOD natural-language experiment using Llama-3-8b, demonstrating that ICL tends to make in-distribution predictions.
   * In Sec. 4.3, we add a novel natural-language classification experiment using Llama-3-8b, showing that ICL can implement algorithm selection based on the low-test-error preference.
   * In Sec 3.3, we rerun the OOD synthetic word classification task using Llama-3-8b, revealing the limitation of ICL in learning new task functions again.
   * In Appendix B.2, we we train a GPT-2 to perform the same synthetic task as in Section 3.3 to show that the task is easy to solve when it's in-distribution, which demonstrates the poor performance mainly stems from the OOD nature of the task.
2. Improve the presentations. To improve clarity, we refine the presentation of the following sections:
   * In Sec. 3.1, clarify the setup and the observations of the retrieval task in Fig. 4
   * In Sec. 3.2, clarify the motivation and the observations of the OOD version of the predict-then-retrieve task in Fig. 6
   * In Sec. 3.3, clarify the setup of the retrieval version of the synthetic OOD classification task in Fig. 7
   * In Sec. 3.3, add some insights for practical applications of real LLMs
3. Add more discussions.
   * In Appendix A.3, we add discussions about previous works on the Bayesian-optimal framework for understanding ICL. We discussion the limitations of previous works and clarify how our work improves them.

---

### Meta-Review · Area_Chair_dn3e · 2024-12-08

**Metareview:**

This paper investigates the behavior of in-context learning (ICL) on out-of-distribution tasks through empirical analysis and theory development. The reviewers generally found merit in the paper's experimental methodology and results, particularly highlighting the well-designed synthetic tasks and comprehensive evaluation approach. The paper received ratings ranging from 5 to 8, with reviewers acknowledging both its empirical contributions in characterizing ICL's behavior on OOD tasks and its theoretical insights into how ICL implements functions from its pretraining distribution, though half of the reviewers found the presentation not unified and one felt this paper was subsumed by existing Bayesian ICL theories. Nevertheless, with an appropriate revision for clarity and comparison with the Bayesian framing of ICL which is prevalant, I assess that this work would be a good fit for this ICLR.

**Additional Comments On Reviewer Discussion:**

A central point of discussion in the review process concerned the theoretical framing of the results, particularly regarding Bayesian interpretations of ICL. Following reviewer feedback, the authors added discussion of the Bayesian perspective in the supplementary materials. To further strengthen the manuscript, incorporating some of this theoretical context into the main text would help readers better understand the empirical findings. As this revision would not fundamentally alter the paper's core contributions, and given the authors' constructive engagement with reviewer feedback, the paper appears suitable for inclusion in ICLR.

---

### Decision · Program_Chairs · 2025-01-22

Accept (Poster)